# Learning Human-Like RL Agents through Trajectory Optimization with Action Quantization

**Jian-Ting Guo[1]\***, **Yu-Cheng Chen[1]\***, **Ping-Chun Hsieh[1]†**, **Kuo-Hao Ho[1]**
**Po-Wei Huang[1]**, **Ti-Rong Wu[2]†**, **I-Chen Wu[1,3]**

[1]Department of Computer Science, National Yang Ming Chiao Tung University, Taiwan
[2]Institute of Information Science, Academia Sinica, Taiwan
[3]Research Center for Information Technology Innovation, Academia Sinica, Taiwan

{gjt.cs13, yucheng.cs11, pinghsieh, lukewayne123.cs05}@nycu.edu.tw
a311551048.cs11@nycu.edu.tw, tirongwu@iis.sinica.edu.tw, icwu@cs.nycu.edu.tw

## Abstract

Human-like agents have long been one of the goals in pursuing artificial intelligence. Although reinforcement learning (RL) has achieved superhuman performance in many domains, relatively little attention has been focused on designing human-like RL agents. As a result, many reward-driven RL agents often exhibit unnatural behaviors compared to humans, raising concerns for both interpretability and trustworthiness. To achieve human-like behavior in RL, this paper first formulates human-likeness as trajectory optimization, where the objective is to find an action sequence that closely aligns with human behavior while also maximizing rewards, and adapts the classic receding-horizon control to human-like learning as a tractable and efficient implementation. To achieve this, we introduce Macro Action Quantization (MAQ), a human-like RL framework that distills human demonstrations into macro actions via Vector-Quantized VAE. Experiments on D4RL Adroit benchmarks show that MAQ significantly improves human-likeness, increasing trajectory similarity scores, and achieving the highest human-likeness rankings among all RL agents in the human evaluation study. Our results also demonstrate that MAQ can be easily integrated into various off-the-shelf RL algorithms, opening a promising direction for learning human-like RL agents. Our code is available at https://rlg.iis.sinica.edu.tw/papers/MAQ.

## 1 Introduction

Designing human-like agents has been an important goal on the path toward artificial intelligence. One of the most well-known benchmarks is the *Turing Test*, which evaluates whether an agent can perform intelligent behavior that is indistinguishable from a human. To pass the Turing Test, agents are designed not only to achieve the goal of the given task but also to behave in a human-like manner.

While the pursuit of human-like agents has been investigated in natural language processing (NLP), such as in large language models (LLMs) that aim to generate responses aligned with human preferences [1], it has not yet been widely explored in the field of deep reinforcement learning (DRL). In recent years, DRL has made significant progress in various domains, such as gaming [2, 3, 4] or robotics control [5, 6]. However, researchers have noticed RL agents often exhibit unnatural behaviors,

---

*These authors contributed equally.
†Corresponding authors.

39th Conference on Neural Information Processing Systems (NeurIPS 2025).

making they easily distinguishable from those of human players, e.g., smoothness [7, 8, 9, 10], navigation issues [11, 12, 13], and (unnatural) shaking and spinning actions [14]. Although RL agents can accomplish these tasks, they fail to achieve the goal of designing human-like intelligence. This disparity raises an important and underexplored challenge in human-like RL research: *How to design agents that not only succeed in tasks but also behave like humans?*

To address this challenge, this paper proposes to formulate human-likeness in reinforcement learning as a trajectory optimization problem, where the objective is to produce trajectories that closely align with human behavior. Following the principle of receding-horizon control, we optimize over short action segments rather than entire trajectories to capture human behavior more effectively. To achieve this, we introduce *Macro Action Quantization* (MAQ), a human-likeness aware RL framework for developing competitive agents with human-like behavior. MAQ first distills human behavior into *macro actions* – sequences of actions – from offline human demonstrations using a conditional Vector-Quantized Variational Autoencoder (VQVAE) [15]. The VQVAE generates a discrete set of macro actions, stored in a learned *codebook*, where each entry represents possible human behavior segments for a given state. By leveraging these macro actions, MAQ transforms the action space from low-level primitive actions to high-level, human-like macro actions, effectively constraining the agent to operate within a human-like behavioral space.

We evaluate MAQ on one of the standard RL benchmarks, the Adroit tasks in D4RL [16], which has readily available human demonstrations. We apply MAQ to three RL algorithms, including IQL, SAC, and RLPD. Experimental results show that MAQ not only achieves higher success rates in task completion but also substantially improves human-likeness. When measuring trajectory similarity using Dynamic Time Warping (DTW) and Wasserstein distance metrics, MAQ significantly increases the similarity scores across all tasks. In addition, we conduct a human evaluation study – similar to the Turing Test – in which participants are asked to distinguish between human demonstrations and agent behaviors. The results show that while participants can easily identify traditional RL agents as non-human, they struggle to distinguish MAQ agents from humans. In conclusion, our findings demonstrate that MAQ effectively captures human-like behavior, offering a promising direction for future research in human-like RL studies.

## 2 Background

### 2.1 Human-like Reinforcement Learning

While most RL research focuses on designing reward-driven RL agents, there are a few works exploring human-like RL in two directions. The first involves adopting *behavioral constraints* to enforce human-likeness. For example, Fujii et al. [17] proposed to penalize actions identified as non-human, while Ho et al. [14] introduced the adaptive behavior cost to discourage non-human-like behavior, such as spinning and shaking. Although effective, these methods rely on pre-defined behavior constraints or rule-based penalties, requiring substantial effort for handcrafted design. An alternative approach to avoid the handcrafted design of behavior costs is to directly learn from *offline human datasets*. Notably, human demonstrations are available and widely used for RL training in various domains, such as Atari games [18], self-driving car [19], and robot arm manipulation [16]. These pre-collected demonstrations can be used to capture the characteristics of human behavior through imitation learning [20], such as behavior cloning or inverse RL [21]. However, while imitation learning improves human likeness, its performance is often limited by the quality of human demonstrations and fails to compete with the non-human-like RL benchmarks. In summary, how to effectively train human-like RL agents remains an open and underexplored challenge.

### 2.2 Semi-Markov Decision Process and Macro Action

Semi-Markov Decision Process (SMDP) extends MDP by incorporating *macro actions*, which are sequences of consecutive actions executed over multiple timesteps. Given a state $s_t$, a macro action with a length $L$ is defined as $m_t = (a_t, a_{t+1}, \ldots, a_{t+L-1})$. Compared to MDP, SMDP is denoted by $(\mathcal{S}, \mathcal{M}, \mathcal{R}, \mathcal{P}, \gamma)$, where $\mathcal{M}$ is the set of macro actions and $\mathcal{R}$ is the cumulative reward obtained over the macro actions: $R(s_t, m_t) = \Sigma_{k=t}^{t+L-1} r(s_k, a_k)$ [22]. Similarly, the objective is to learn a policy that selects macro actions to maximize the cumulative rewards: $J(\pi_\theta) = \mathbb{E}_{(s_t, m_t) \sim \pi_\theta}[\Sigma_{t=0}^{\infty} \gamma^t R(s_t, m_t)]$, where $m_t \in M$, and $t$ represents the timestep for selecting macro

actions. SMDP simplifies complex tasks by enabling agents to focus on long-term planning, as macro actions reduce decision-making frequency while capturing sequential dependencies.

## 2.3 VQVAE

*Vector quantized variational autoencoder* (VQVAE) [15], an extension of the *variational autoencoder* (VAE) [23], is a generative model that incorporates vector quantization to learn discrete latent embeddings. VQVAE consists of an encoder, a codebook of discrete latent embeddings, and a decoder. The encoder transforms input data into a latent vector, which is then quantized by replacing it with the nearest vectors from the codebook. The decoder aims to reconstruct the same input data using this discrete latent code. By utilizing discrete latent representations, VQVAE allows learning more structured representations.

Recent studies have explored the application of VQVAE in reinforcement learning. For example, Ozair et al. [24] and Antonoglou et al. [25] propose using conditional-VQVAE [26] as a state transition model in stochastic environments, using the codebook's latent vectors to represent possible chance events. Given a state and a latent embedding (chance event) from the codebook, the decoder aims to generate the corresponding next states. In addition, Luo et al. [27] proposes utilizing conditional-VQVAE to discretize continuous action spaces in robotic control tasks. Specifically, each latent embedding in the codebook represents a possible action for a given state. The decoder reconstructs the action from the corresponding latent embedding. This approach transforms continuous action spaces into discrete action spaces, improving learning efficiency in robotic control tasks.

# 3 Human-Like Trajectory Optimization

**Trajectory optimization.** The canonical trajectory optimization problem is formulated based on an MDP $(\mathcal{S}, \mathcal{A}, \mathcal{P}, R, \gamma, T, p_0)$ with continuous state space $\mathcal{S}$ and action space $\mathcal{A}$, transition function $\mathcal{P} : \mathcal{S} \times \mathcal{A} \to \Delta(\mathcal{S})^3$, reward function $R : \mathcal{S} \times \mathcal{A} \to \mathbb{R}$, discount factor $\gamma \in [0, 1)$, episode length $T$, and initial state distribution $p_0$. Let $\tau = (s_0, a_0, \cdots, s_{T-1}, a_{T-1}, s_T)$ denote a trajectory generated under an action sequence $a_{0:T-1} := (a_0, a_1, \cdots, a_{T-1})$. Our goal is to find an action sequence that can maximize the total discounted return.

$$a_{0:T-1}^* := \arg\max_{a_{0:T-1} \in \mathcal{A}^T} \mathbb{E}\left[ \sum_{t=0}^{T-1} \gamma^t R(s_t, a_t) \Big| s_0 \sim p_0; a_{0:T-1} \right], \tag{1}$$

where the expectation is taken over the randomness of the initial state, reward realizations, and state transitions.

**Human-like receding-horizon control (HRC).** As exact full-horizon optimization in equation (1) can be intractable even for moderately large $T$, we resort to the technique of receding-horizon control, which is a generic control method that involves repeatedly solving a optimization problem on a short moving time horizon to choose action sequences. Specifically, at the decision step $t$, receding-horizon control would find an action sequence of length $H$ (usually much smaller than $T$) by solving

$$\arg\max_{a_{t:t+H-1}} \mathbb{E}\left[ \sum_{i=t}^{t+H-1} \gamma^{i-t} R(s_i, a_i) \Big| s_t; a_{t:t+H-1} \right]. \tag{2}$$

Notably, under receding-horizon control, such short-term replanning is done every $k$ steps (with $1 \le k \le H$), *i.e.*, the first $j$ actions in $a_{t:t+H-1}^*$ ($1 \le j \le H$) are executed and the environment returns $s_{t+j}$ and equation (2) is re-optimized. This *receding-horizon* loop reproduces the closed-loop robustness observed in [28] while bounding per-step computation.

To enforce human-likeness, we introduce the concept of *human-like sequence constraint* as follows: Let $\mathcal{D} = \{\tau^{(i)}\}_{i=1}^N$ be a set of human-generated action sequences and define *human manifold* $\mathcal{H} = \{\tau \mid \tau \in \mathcal{D}\}$. Accordingly, human-likeness can be enforced by constraining the search space of action sequences to $\mathcal{H}$. Hence, human-like receding-horizon control can be formally described as finding an action sequence as

$$a_{t:t+H-1}^* = \arg\max_{a_{t:t+H-1} \in \mathcal{H}} \mathbb{E}\left[ \sum_{i=t}^{t+H-1} \gamma^{i-t} R(s_i, a_i) \Big| s_t; a_{t:t+H-1} \right]. \tag{3}$$

---

[3]Throughout this paper, we use $\Delta(\mathcal{X})$ to denote the set of all probability distributions over a set $\mathcal{X}$.

Under HRC, executing a larger prefix $j$ of the $H$-step action sequence generates longer, uninterrupted human-style motion, whereas $j = 1$ maximizes reactivity.

**Action-segment quantization for efficient HRC.** Implementing equation (3) presents two obstacles: (1) Searching for all $|\mathcal{H}|$ demonstration segments at every step is prohibitively slow. (2) Although the dataset records each visited state, the starting state of a stored segment still rarely *matches the* current *state $s_t$ exactly*, so naively re-using segments requires expensive alignment or local optimization. To alleviate these issues, we propose to perform *segment-level planning*: HRC first proposes a complete $H$-step sequence $(a_t, a_{t+1}, \ldots, a_{t+H-1})$, executes *all $H$* actions, observes the resulting state $s_{t+H}$, and then replans a *new $H$-step* action sequence from that state. This commit-and-replan loop amortizes the optimization overhead over fixed-length segments while fully re-using each trajectory rollout.

Moreover, searching for this segment in continuous action space is still computationally very costly, so we propose to adopt the *macro action quantization* strategy. Inspired by [27], we replace each segment of continuous actions with a compact sequence of discrete codebook indices, effectively reducing the search over the full space $\mathcal{A}^H$ to a lookup on a finite codebook. Since all demonstrations can be rolled out and scored *offline*, the HRC controller needs only a constant-time table lookup at runtime, slashing planning latency while still retaining the multi-step look-ahead and human-style fidelity provided by $H$-step segments.

## 4 Macro Action Quantization

To concretize the trajectory optimization introduced in Section 3, we propose a human-likeness aware framework called *Macro Action Quantization* (MAQ). MAQ addresses the challenges of enforcing human-likeness in receding-horizon control by learning a quantized set of action segments – macro actions – from offline human demonstrations. These macro actions serve as $H$-step segments in the trajectory optimization process. The primary goal of MAQ is to achieve high performance while preserving the key characteristics of human-like behavior. We present the details of the MAQ framework as follows.

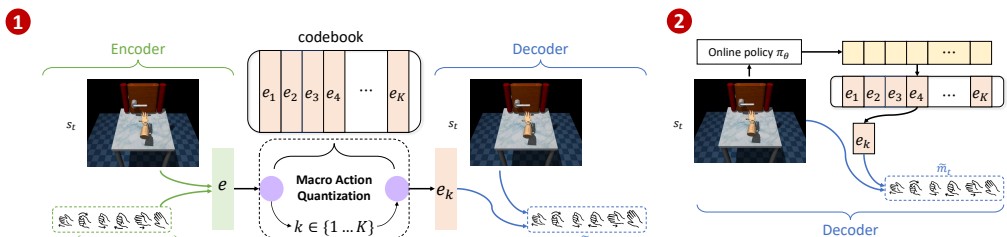

Figure 1: The overall training process of MAQ. (1) **Human behavior distillation**: A Conditional-VQVAE is trained on the state ($s_t$) and the macro action ($m_t = (a_t, a_{t+1}, \ldots, a_{t+H-1})$) to learn a discrete codebook; the macro actions ($m_t$) are extracted from human demonstrations via a sliding window over action trajectories. (2) **Reinforcement learning with MAQ**: An online policy ($\pi_\theta$) acts in the learned discrete code space by selecting codebook indices, which the VQVAE decodes into macro action executed in the environment.

**Human behavior distillation.** To capture the characteristics of human behavior, specifically the *action sequence* patterns, we propose training a Conditional-VQVAE to distill macro actions from human demonstrations, as illustrated on the left in Figure 1. Specifically, given a state $s_t$ and a macro action $m_t$, where $m_t = (a_t, a_{t+1}, \ldots, a_{t+H-1})$ represents a sequence of $H$ consecutive actions executed by a human at $s_t$, the encoder generates a latent vector $e$. This vector $e$ is then compared with the codebook to identify the nearest latent vector $e_k$. Finally, $e_k$ along with the state $s_t$ is passed to the decoder to produce a reconstructed macro action $\widetilde{m}_t$, which aims to replicate the original macro action $m_t$. During training, the loss function is updated as follows:

$$\mathcal{L} = ||m - \widetilde{m}||^2 + ||sg[e] - e_k||^2 + \beta ||e - sg[e_k]||^2, \tag{4}$$

where $sg$ denotes the stop-gradient operation and $\beta$ is the coefficient of commitment loss. MAQ not only captures the human behavior of a specific state and macro action pair $(s_t, m_t)$ but also

generalizes the macro actions across similar states, with the codebook serving as a collection of human-like behaviors mapped to different states.

MAQ significantly reduces the macro action space. Since macro actions consist of sequences of actions, the number of possible macro actions grows exponentially with the sequence length. For example, given a sequence length $H$ and assuming each state has $n$ potential primitive actions, there are $n^H$ possible macro actions. Fortunately, as the goal of MAQ is to capture human-like behavior, a vast number of these macro actions can be disregarded in practice since they do not appear in human demonstration. Therefore, we limit the codebook size to $K$, focusing only on macro actions that align with human behavior, effectively reducing the action space and distilling the key macro actions.

**Reinforcement learning with MAQ.** MAQ can be seamlessly integrated into any reinforcement learning algorithm. Conceptually, by quantizing macro actions, the action space is transformed from primitive actions (MDP) to macro actions (SMDP), while all other aspects of the RL training remain unchanged. Since these macro actions are distilled from human demonstration, RL agents aim to achieve high performance while operating within the constraints of human-like behaviors.

Specifically, we train a policy $\pi_\theta$ to select the index $k$ corresponding to the codebook entry. The policy $\pi_\theta$ takes a given state $s$ as input and produces a distribution with $K$ policy logits, each corresponding to one macro action embedding in the codebook. At each state $s_t$, $\pi_\theta$ samples an index according to the policy distribution. Suppose the index $k$ is selected, the corresponding latent vector $e_k$ is then retrieved from the codebook. With this latent vector $e_k$ and the state $s_t$, the decoder reconstructs a macro action $\widetilde{m}_t$ to interact with the environment, as illustrated on the right in Figure 1.

To optimize $\pi_\theta$, we follow equation (3) to find a macro action that maximizes the expected reward

$$m_t^\star = \arg\max_{m_t \in \mathcal{H}} \mathbb{E}\left[R(s_t, m_t)\Big| s_t; m_t\right], \tag{5}$$

where $m_t = a_{t:t+H-1}$ is the macro action and $R(s_t, m_t) = \sum_{i=t}^{t+H-1} \gamma^{i-t} R(s_i, a_i)$ is the cumulative reward over the segment. Through this process, MAQ allows agents to plan and act with human-like behavior while optimizing for long-term return.

# 5 Experiments

This section empirically evaluates the proposed MAQ framework. Our experiments are designed to answer the two main questions: *Q1) Can MAQ accomplish tasks while optimizing trajectories that align with human demonstrations*, and *Q2) Can MAQ exhibit human-like behavior and convince human evaluators into believing it is human in a Turing Test evaluation?*

## 5.1 Experiment Setup

Our experiments are conducted on four Adroit tasks from D4RL [16], including *Door* (opening a door), *Hammer* (hammering a nail), *Pen* (twirling a pen), and *Relocate* (lifting and moving a ball). We first train three off-the-shelf RL algorithms, including IQL [29], SAC [30], and the state-of-the-art offline algorithm RLPD [6]. Next, we apply MAQ to each of these algorithms, resulting in three human-like RL agents – MAQ+IQL, MAQ+SAC, and MAQ+RLPD – to evaluate their generality across different RL algorithms. When incorporating MAQ, each RL algorithm learns a policy over a discrete codebook, replacing the original primitive action space with macro actions distilled from human demonstrations. This flexibility makes MAQ easy to apply to a wide range of RL algorithms. Note that to support the discrete action space introduced by MAQ, we adopt Discrete SAC (DSAC) [31] when applying it to SAC. In addition, we train a Behavioral Cloning (BC) [32] agent as a baseline for evaluating human-likeness.

Each task includes 25 human demonstration trajectories, which are split into training and testing datasets with a 9:1 ratio. Agents that require human demonstrations, including BC, IQL, and all MAQ-based agents, are trained on the same training datasets. All agents are evaluated based on trajectory similarity on the testing dataset. To ensure a fair comparison, all agents (including baseline and MAQ-based agents) are trained for a total of $10^6$ steps. For IQL and MAQ+IQL, which include both offline and online training stages, we train $10^6$ steps to each stage. For MAQ-based agents, we first train a VQVAE on human demonstrations to obtain the macro action codebook. The VQVAE is

trained with $\beta = 0.25$ in Eq. (4), a hidden size of 256, a batch size of 32, and 100 training episodes. The same trained VQVAE is shared across all MAQ-based agents to ensure consistency. Other detailed hyperparameters are provided in Appendix A.

## 5.2 Trajectory Similarity Evaluation

### 5.2.1 Evaluation Metrics

To evaluate how closely agent behaviors align with human demonstrations, we use two trajectory similarity metrics: Dynamic Time Warping (DTW) and the Wasserstein Distance (WD). DTW measures the alignment between trajectories, while WD captures distributional similarity. Both metrics are computed between agent and human trajectories from the same testing dataset.

**Dynamic time warping.** DTW is a technique used to measure the similarity between two sequences, even if they differ in length or timing. It finds the optimal alignment between two sequences by stretching or compressing segments to minimize the overall distance. Generally, a lower distance indicates higher similarity. Specifically, let $\mathcal{T}^H = \{\tau_1^H, \tau_2^H, \ldots, \tau_n^H\}$ and $\mathcal{T}^A = \{\tau_1^A, \tau_2^A, \ldots, \tau_m^A\}$ denote the sets of trajectories played by the human and the agent, consisting of $n$ and $m$ trajectories, respectively. The DTW distance between $\mathcal{T}^H$ and $\mathcal{T}^A$ is calculated by $\text{DTW} = \frac{1}{n} \sum_{i=1}^{n} \left( \frac{1}{m} \sum_{j=1}^{m} DTW(\tau^H, \tau^A) \right)$. Moreover, we compare two types of DTW distances: state distance $\text{DTW}_s$ and action distance $\text{DTW}_a$. The $\text{DTW}_s$ and $\text{DTW}_a$ are calculated by applying DTW to the state and action sequences in $DTW(\tau^H, \tau^A)$, respectively. A lower $\text{DTW}_s$ indicates that the agent frequently encounters states in a similar order to those in human demonstrations, while a lower $\text{DTW}_a$ suggests that the agent performs actions in an order similar to human demonstrations.

**Wasserstein distance.** Compared to DTW, WD evaluates distributional similarity with the Wasserstein distance between the agent and human demonstrations: $W(\rho_{\text{agent}}, \rho_{\text{human}})$. The calculation is performed on the same normalized feature space used for DTW and uses the POT library's `emd2` solver to compute the Wasserstein distance. Similar to DTW, we evaluate two Wasserstein variants: state-based Wasserstein $\text{WD}_s$ and action-based Wasserstein $\text{WD}_a$. The $\text{WD}_s$ and $\text{WD}_a$ are calculated by applying $W(\rho_{\text{agent}}, \rho_{\text{human}})$ to the state and action.

### 5.2.2 Performance of MAQ-based Agents

Table 1 summarizes results for the four Adroit tasks, reporting state and action-based Dynamic Time Warping ($\text{DTW}_s$, $\text{DTW}_a$) and Wasserstein distance ($\text{WD}_s$, $\text{WD}_a$) scores together with task success rates. For better comparability across tasks, we normalize the similarity scores for each task using $1 - \frac{\text{agent score} - \text{human score}}{\text{random score} - \text{human score}}$, where the agent score is the similarity between the agent and human trajectories, the human score is the similarity between human trajectories, and the random score is the similarity between a random agent and human trajectories. After transformation, higher values indicate more human-like behavior.

Table 1 shows that incorporating MAQ significantly increases trajectory similarity across all tasks, according to both DTW and WD scores. For example, in the *Door* task, $\text{DTW}_s$ improves from 0.43 with IQL to 0.84 with MAQ+IQL, from -0.39 with SAC to 0.8 with MAQ+SAC, and from -0.06 with RLPD to 0.76 with MAQ+RLPD. Similarly, in the *Hammer* task, $\text{WD}_a$ improves from -0.19 with SAC to 0.78 with MAQ+SAC, and from 0.2 with RLPD to 0.85 with MAQ+RLPD. Most importantly, these improvements in trajectory similarity are achieved without significantly sacrificing task success rate. In addition, although the BC agent learns directly from human demonstrations and achieves moderate similarity scores, it performs the worst in terms of task success rate among all agents. This is because the BC agent learns to imitate state-action pairs from the demonstrations, without optimizing for successful outcomes. Overall, the largest improvement is observed between SAC and MAQ+SAC, where the average $\text{DTW}_s$ increases from -0.49 to 0.56. This is because SAC learns solely from environment interaction without using any human demonstrations, unlike IQL and RLPD. As a result, its trajectories deviate more significantly from those of humans. Our findings also suggest that MAQ has significant potential when applied to more complex problems, such as tasks requiring intricate scenarios with numerous macro actions (e.g., continuous action spaces).

Table 1: The results for Adroit benchmark from D4RL. The baseline of each algorithm incorporate with MAQ generally improves all of the original algorithm similarity score in each control task.

| Tasks | | BC | IQL | MAQ+IQL | SAC | MAQ+SAC | RLPD | MAQ+RLPD |
|---|---|---|---|---|---|---|---|---|
| Door | DTW$_s$($\uparrow$) | $0.18 \pm 0.09$ | $0.43 \pm 0.06$ | $\mathbf{0.84 \pm 0.06}$ | $-0.39 \pm 0.10$ | $\mathbf{0.80 \pm 0.08}$ | $-0.06 \pm 0.04$ | $\mathbf{0.76 \pm 0.04}$ |
| | DTW$_a$($\uparrow$) | $0.42 \pm 0.13$ | $0.61 \pm 0.04$ | $\mathbf{0.95 \pm 0.01}$ | $-0.25 \pm 0.04$ | $\mathbf{0.91 \pm 0.03}$ | $0.28 \pm 0.08$ | $\mathbf{0.91 \pm 0.05}$ |
| | WD$_s$($\uparrow$) | $0.32 \pm 0.05$ | $0.48 \pm 0.05$ | $\mathbf{0.75 \pm 0.05}$ | $-0.28 \pm 0.02$ | $\mathbf{0.71 \pm 0.08}$ | $-0.14 \pm 0.04$ | $\mathbf{0.71 \pm 0.03}$ |
| | WD$_a$($\uparrow$) | $0.41 \pm 0.08$ | $0.50 \pm 0.02$ | $\mathbf{0.81 \pm 0.03}$ | $-0.15 \pm 0.02$ | $\mathbf{0.77 \pm 0.07}$ | $0.10 \pm 0.02$ | $\mathbf{0.76 \pm 0.03}$ |
| | Success($\uparrow$) | $0.02 \pm 0.01$ | $0.16 \pm 0.06$ | $\mathbf{0.93 \pm 0.04}$ | $0.43 \pm 0.23$ | $\mathbf{0.56 \pm 0.50}$ | $0.96 \pm 0.07$ | $0.93 \pm 0.05$ |
| Hammer | DTW$_s$($\uparrow$) | $-0.16 \pm 0.07$ | $-0.14 \pm 0.34$ | $\mathbf{0.64 \pm 0.17}$ | $-1.10 \pm 0.35$ | $\mathbf{0.61 \pm 0.21}$ | $-0.03 \pm 0.14$ | $\mathbf{0.68 \pm 0.17}$ |
| | DTW$_a$($\uparrow$) | $0.47 \pm 0.02$ | $0.45 \pm 0.20$ | $\mathbf{0.92 \pm 0.06}$ | $-0.33 \pm 0.11$ | $\mathbf{0.91 \pm 0.10}$ | $0.37 \pm 0.08$ | $\mathbf{0.94 \pm 0.07}$ |
| | WD$_s$($\uparrow$) | $0.11 \pm 0.06$ | $0.12 \pm 0.13$ | $\mathbf{0.75 \pm 0.03}$ | $-0.44 \pm 0.09$ | $\mathbf{0.64 \pm 0.12}$ | $-0.03 \pm 0.08$ | $\mathbf{0.76 \pm 0.04}$ |
| | WD$_a$($\uparrow$) | $0.30 \pm 0.03$ | $0.30 \pm 0.10$ | $\mathbf{0.84 \pm 0.02}$ | $-0.19 \pm 0.04$ | $\mathbf{0.78 \pm 0.11}$ | $0.20 \pm 0.03$ | $\mathbf{0.85 \pm 0.03}$ |
| | Success($\uparrow$) | $0.00 \pm 0.00$ | $\mathbf{0.01 \pm 0.01}$ | $0.00 \pm 0.00$ | $\mathbf{0.01 \pm 0.01}$ | $0.00 \pm 0.00$ | $\mathbf{1.00 \pm 0.00}$ | $0.56 \pm 0.37$ |
| Pen | DTW$_s$($\uparrow$) | $0.53 \pm 0.13$ | $0.34 \pm 0.09$ | $\mathbf{0.55 \pm 0.17}$ | $0.06 \pm 0.20$ | $\mathbf{0.58 \pm 0.17}$ | $0.48 \pm 0.24$ | $\mathbf{0.54 \pm 0.18}$ |
| | DTW$_a$($\uparrow$) | $0.58 \pm 0.05$ | $0.51 \pm 0.05$ | $\mathbf{0.58 \pm 0.09}$ | $-0.34 \pm 0.16$ | $\mathbf{0.58 \pm 0.11}$ | $0.40 \pm 0.17$ | $\mathbf{0.59 \pm 0.13}$ |
| | WD$_s$($\uparrow$) | $0.59 \pm 0.12$ | $0.54 \pm 0.11$ | $\mathbf{0.59 \pm 0.13}$ | $0.29 \pm 0.10$ | $\mathbf{0.61 \pm 0.12}$ | $0.49 \pm 0.08$ | $\mathbf{0.59 \pm 0.12}$ |
| | WD$_a$($\uparrow$) | $0.65 \pm 0.14$ | $0.63 \pm 0.12$ | $\mathbf{0.66 \pm 0.15}$ | $0.22 \pm 0.15$ | $\mathbf{0.67 \pm 0.14}$ | $0.44 \pm 0.12$ | $\mathbf{0.66 \pm 0.14}$ |
| | Success($\uparrow$) | $0.40 \pm 0.03$ | $0.40 \pm 0.05$ | $\mathbf{0.42 \pm 0.07}$ | $0.32 \pm 0.09$ | $\mathbf{0.41 \pm 0.01}$ | $0.62 \pm 0.09$ | $0.42 \pm 0.05$ |
| Relocate | DTW$_s$($\uparrow$) | $0.09 \pm 0.14$ | $0.20 \pm 0.20$ | $\mathbf{0.52 \pm 0.06}$ | $-0.55 \pm 0.20$ | $\mathbf{0.25 \pm 0.19}$ | $0.03 \pm 0.13$ | $\mathbf{0.27 \pm 0.14}$ |
| | DTW$_a$($\uparrow$) | $0.47 \pm 0.15$ | $0.51 \pm 0.11$ | $\mathbf{0.82 \pm 0.01}$ | $-0.10 \pm 0.16$ | $\mathbf{0.66 \pm 0.09}$ | $0.32 \pm 0.13$ | $\mathbf{0.69 \pm 0.10}$ |
| | WD$_s$($\uparrow$) | $0.27 \pm 0.15$ | $0.36 \pm 0.06$ | $\mathbf{0.47 \pm 0.07}$ | $-0.22 \pm 0.06$ | $\mathbf{0.40 \pm 0.07}$ | $0.02 \pm 0.07$ | $\mathbf{0.38 \pm 0.09}$ |
| | WD$_a$($\uparrow$) | $0.45 \pm 0.12$ | $0.50 \pm 0.04$ | $\mathbf{0.65 \pm 0.03}$ | $-0.05 \pm 0.03$ | $\mathbf{0.61 \pm 0.03}$ | $0.20 \pm 0.03$ | $\mathbf{0.55 \pm 0.08}$ |
| | Success($\uparrow$) | $0.01 \pm 0.02$ | $0.00 \pm 0.00$ | $\mathbf{0.20 \pm 0.10}$ | $0.00 \pm 0.00$ | $\mathbf{0.14 \pm 0.07}$ | $0.14 \pm 0.03$ | $\mathbf{0.17 \pm 0.10}$ |
| Average | DTW$_s$($\uparrow$) | $0.16 \pm 0.29$ | $0.21 \pm 0.25$ | $\mathbf{0.63 \pm 0.14}$ | $-0.49 \pm 0.48$ | $\mathbf{0.56 \pm 0.23}$ | $0.10 \pm 0.25$ | $\mathbf{0.56 \pm 0.21}$ |
| | DTW$_a$($\uparrow$) | $0.49 \pm 0.07$ | $0.52 \pm 0.06$ | $\mathbf{0.82 \pm 0.17}$ | $-0.26 \pm 0.11$ | $\mathbf{0.76 \pm 0.17}$ | $0.34 \pm 0.05$ | $\mathbf{0.78 \pm 0.17}$ |
| | WD$_s$($\uparrow$) | $0.32 \pm 0.20$ | $0.38 \pm 0.18$ | $\mathbf{0.64 \pm 0.14}$ | $-0.17 \pm 0.32$ | $\mathbf{0.59 \pm 0.14}$ | $0.08 \pm 0.28$ | $\mathbf{0.61 \pm 0.17}$ |
| | WD$_a$($\uparrow$) | $0.45 \pm 0.15$ | $0.48 \pm 0.14$ | $\mathbf{0.74 \pm 0.10}$ | $-0.04 \pm 0.18$ | $\mathbf{0.71 \pm 0.08}$ | $0.24 \pm 0.15$ | $\mathbf{0.70 \pm 0.13}$ |
| | Success($\uparrow$) | $0.11 \pm 0.19$ | $0.14 \pm 0.19$ | $\mathbf{0.39 \pm 0.40}$ | $0.19 \pm 0.22$ | $\mathbf{0.28 \pm 0.25}$ | $\mathbf{0.68 \pm 0.40}$ | $0.52 \pm 0.32$ |

### 5.2.3 Human-Like Trajectory Optimization with Different Macro Action Lengths

We examine how the macro action length $H$ in the MAQ codebook affects the human-likeness similarity score. We evaluate the MAQ+RLPD across different sequence lengths ($H = 1$ to $9$), using the same five metrics illustrated in Table 1, averaged over the four *Adroit* tasks. Figure 2 shows the results, where each cell is color-coded with darker shades indicating higher values. The color scale is normalized independently for each metric. The results show that $H = 9$ yields the highest similarity scores as well as the best task success rate. Notably, WD$_a$ remains relatively stable across all values of $H$, likely due to the nature of the Adroit action space, which consists solely of Shadow Hand joint positions. This suggests that even shorter sequences can appear human-like in action space without being effective at completing the task. As $H$ increases, both trajectory similarity and success rate improve, indicating that longer macro actions not only enhance human-likeness similarity score but also contribute to more effective task execution, highlighting the benefits of temporally extended planning in human-like trajectory optimization. We have also provided additional analysis of MAQ in Appendix B.

### 5.3 Human Evaluation Study

While the trajectory similarity scores demonstrate that MAQ-based agents align more closely with human behavior compared to other methods, we also conduct a human evaluation study with 19 human evaluators to verify whether their behavior is recognized as human-like by human evaluators. Specifically, each evaluator study is asked to complete two sets of questions, including a Turing Test and a human-likeness ranking test. Detailed settings for human evaluation stduy are provided in Appendix C. The following subsections provide the results on each evaluation test.

### 5.3.1 Turing Test

In the Turing Test, we conduct several two-alternative forced-choice (2AFC) questions, where evaluators are shown two videos – one from human demonstrations and the other from one of the seven trained agents listed in Table 1 – and asked to choose which one was performed by humans. To ensure fairness, the evaluation consists of seven questions for each Adroit task, each corresponding to a different agent. Namely, each agent appears exactly once, and the order of appearance is randomized for each evaluator.

| Sequence Length | DTW$_s$ | DTW$_a$ | WD$_s$ | WD$_a$ | Success |
|---|---|---|---|---|---|
| 9 | 0.56±0.21 | 0.78±0.17 | 0.61±0.17 | 0.70±0.13 | 0.52±0.32 |
| 8 | 0.51±0.24 | 0.77±0.19 | 0.56±0.19 | 0.69±0.14 | 0.42±0.32 |
| 7 | 0.54±0.19 | 0.77±0.18 | 0.58±0.13 | 0.69±0.13 | 0.47±0.30 |
| 6 | 0.57±0.20 | 0.80±0.16 | 0.60±0.15 | 0.71±0.12 | 0.40±0.27 |
| 5 | 0.52±0.22 | 0.76±0.20 | 0.57±0.15 | 0.69±0.13 | 0.37±0.38 |
| 4 | 0.47±0.28 | 0.72±0.19 | 0.54±0.16 | 0.65±0.13 | 0.42±0.38 |
| 3 | 0.46±0.26 | 0.76±0.17 | 0.52±0.17 | 0.68±0.12 | 0.38±0.37 |
| 2 | 0.45±0.23 | 0.73±0.15 | 0.53±0.14 | 0.67±0.10 | 0.30±0.26 |
| 1 | 0.46±0.17 | 0.73±0.17 | 0.54±0.11 | 0.70±0.10 | 0.25±0.25 |

Metrics

Figure 2: Trajectory similarity scores and success rates with different lengths in MAQ+RLPD.

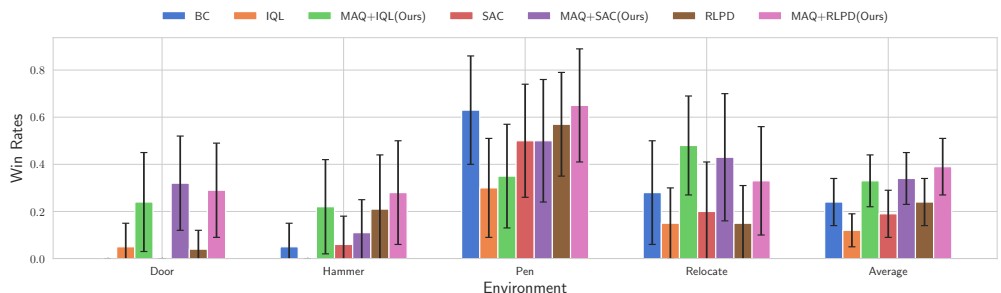

Figure 3: Turing Test. Error bars represent 96% confidence intervals.

Figure 3 shows the results of the Turing Test. Bars indicate the win rate of each agent, i.e., the percentage of questions in which evaluators were fooled by the agent and believed its behavior was more human-like than the human demonstration. The results are consistent with the findings in Subsection 5.2, where MAQ-based agents achieve higher win rates compared to RL algorithms without MAQ. This further corroborates that optimizing trajectories with longer macro action lengths is strongly correlated with human-likeness. Overall, when comparing the average win rate across four tasks, the win rate evaluated by the evaluators is: MAQ+RLPD (39%) > MAQ+SAC (34%) > MAQ+IQL (32%) > BC (24%) = RLPD (24%) > SAC (19%) > IQL (13%). Notably, MAQ+RLPD improves a win rate of 15% higher than non-MAQ agents, demonstrating that our method significantly enhances the perceived human-likeness of agent behavior.

The figure also shows some interesting results. The *Pen* task exhibits the highest win rate for all agents, indicating that evaluators have the greatest difficulty distinguishing agent behavior from human demonstrations in this setting. This suggests that pen manipulating is relatively easy for agents to master, and that the visual differences between human and agent behavior are subtle, making deception more likely. In addition, the SAC agent shows nearly 0% win rates in the *Door* task. Despite achieving a 43% success rate, its behavior remains noticeably different from human demonstrations. This discrepancy clearly demonstrates that reward-driven RL agents are not necessarily human-like.

### 5.3.2 Human-Likeness Ranking Test

The questions in the human-likeness ranking test are similar to those in the Turing Test, but with a key difference, as described below. Since the goal is to compare human-likeness among agents, each question presents evaluators with two videos; however, both videos may be from trained agents.

Evaluators are asked to choose the video that appears more human-like. For fairness, each agent pair appears only once per evaluator, and not all possible agent pairs are shown for evaluators to reduce fatigue. Each evaluator is asked to complete approximately 17 questions during this test. After completing the test, each evaluator is shown two videos – one selected from the most human-like and the other from the least human-like agents or humans based on their responses – and is invited to leave comments explaining their choices.

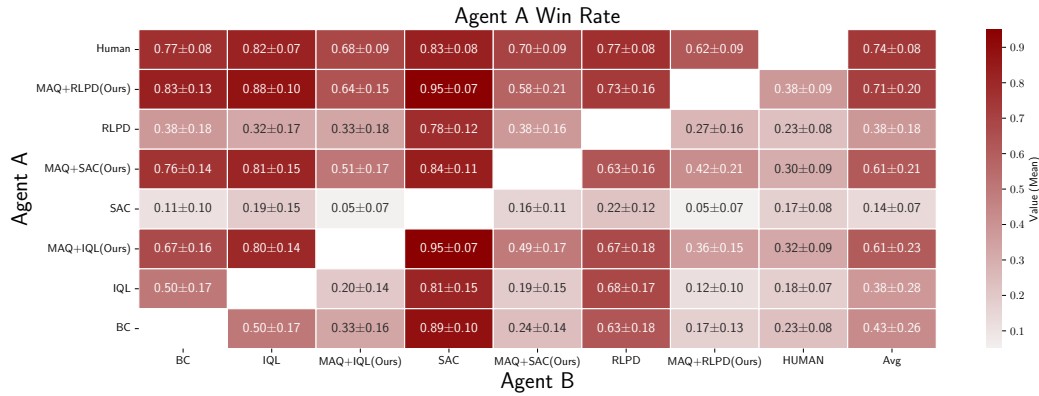

Figure 4: Human-likeness ranking test.

Figure 4 summarizes the results across the four tasks. The heatmap shows the win rate, where each cell indicates how often evaluators judged agents on the y-axis to behave more human-like than agents on the x-axis when comparing their videos directly. Overall, the results are consistent with the Turning Test, where the overall ranking among all agents and humans as follows: Human (74%) > MAQ+RLPD (71%) > MAQ+SAC (61%) = MAQ+IQL (61%) > BC (43%) > RLPD (38%) = IQL (38%) > SAC (14%). Interestingly, the ranking order remains the same as in the Turing Test, except that the positions of IQL and SAC are reversed. This indicates that human evaluators are consistent in their judgments and can distinguish between human behavior and that of RL agents. Moreover, MAQ+RLPD achieves a 71% win rate across all agent pairs, achieving performance comparable to the human demonstration's 74% win rate. This suggests that MAQ+RLPD can convincingly fool human evaluators into believing its behavior is human-like.

### 5.3.3 Human Feedback on Behavior Analysis

We further investigate the feedback provided by human evaluators. Figure 5 shows video clips in *Door*. The MAQ+RLPD agent demonstrates precise control by firmly holding, rotating the door handle, and pulling to open the door, closely mimicking human behavior. In contrast, while the RLPD agent achieves a high success rate in this task, it employs an unnatural method to open the door – using its backhand to press the handle while simultaneously pulling the door without properly holding the handle. Although this approach successfully opens the door, the behavior appears less human-like. Therefore, several evaluators judged the MAQ+RLPD agent to be more human-like. As one evaluator noted, "*Because it shows how a human casually opens a door — the grip on the handle is stable, without forcefully grabbing it all the way*". Conversely, many evaluators believe RLPD agent does not exhibit human-like behavior, remarking, "*Instead of grabbing, it is glitching its hand through the door handle*" and "*Its wrist performed a weird gesture and opened the door rudely.*"

## 6   Discussions

In this paper, we discuss a critical and yet underexplored challenge in reinforcement learning for designing human-like agents. We first formulate human-likness as a trajectory optimization problem, and realize this formulation by proposing MAQ, a human-likeness aware framework that distills human behaviors into macro actions. Our experiments show that MAQ not only completes tasks but also aligns with human behaviors by improving the trajectory similarity scores across four Adroit control tasks, and achieves the highest human-likeness rankings in human evaluation. These results demonstrate that MAQ agents are both effective and convincingly human-like.

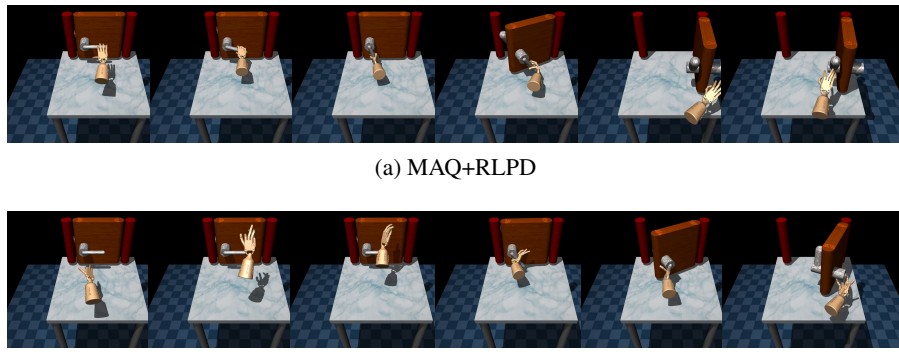

(a) MAQ+RLPD

(b) RLPD

Figure 5: Agent behavior in *Door*.

Although MAQ exhibits more human-like behavior, one limitation is its reliance on human demonstrations to distill macro actions. The quality of these demonstrations can affect the effectiveness of MAQ. However, since our primary goal is to pursue human-likeness, obtaining human demonstrations is both necessary and appropriate within the scope of this work. Moreover, our experiments demonstrate that MAQ can be easily integrated into various RL algorithms. We also show that MAQ has the potential to generalize to more complex domains, such as real-time strategy games, where agents often act at unnaturally high frequencies to maximize performance. Rather than relying on manually designed constraints (e.g., delaying action execution [3]), MAQ naturally regulates decision frequency through learned human-like macro actions, allowing agents to behave more realistically. In conclusion, MAQ offers a promising direction for learning human-like RL agents.

## Broader Impact

This work investigates human-like reinforcement learning by introducing a human-likeness aware framework, called MAQ, that allows AI agents to behave in a more natural and human-like manner. Human-like behavior can offer practical benefits in real-world applications, particularly in scenarios involving human-robot collaboration. For example, a robot that assists with human-like movements can improve safety and trust, reducing the risk of accidents. On the other hand, human-like agents may be misused in deceptive or manipulative ways, such as cheating in competitive games. In summary, the proposed method and human-likeness metrics provide a standardized approach for evaluating and quantifying human-like behavior, benefiting the broader RL community.

## Acknowledgement

This research is partially supported by the National Science and Technology Council (NSTC) of the Republic of China (Taiwan) under Grant Numbers 113-2221-E-001-009-MY3, 113-2634-F-A49-004, 114-2221-E-A49-005, 114-2221-E-A49-006, and 114-2628-E-A49-002. We also thank the National Center for High-performance Computing (NCHC) for providing computational and storage resources. The authors would also like to thank the anonymous reviewers for their valuable comments and the anonymous participants for the Human Evaluation Study.

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

# A  Experimental settings

**Experimental Setup**    We conducted our experiments on a machine equipped with an E5-2678 CPU and four NVIDIA GeForce GTX 1080 Ti GPUs. Training was performed on various games, each requiring a different number of GPU hours depending on the specific task. Our method begins by training a VQVAE with varying sequence lengths $H$ and codebook sizes $K$. For each game, we used three different random seeds: 1, 10, and 100. On average, VQVAE training required less than 5 minutes. Each MAQ-based method is built upon a pre-trained VQVAE. All MAQ variants share the same VQVAE model when trained with the same random seed. We consistently selected the model from the final training iteration as the VQVAE used in our online reinforcement learning (RL) experiments. For online RL training, we evaluated the best performing checkpoint using two similarity metrics: Dynamic Time Warping (DTW) and Wasserstein distance. Among all variants, MAQ+IQL was the most computationally intensive, requiring 9 GPU hours per training run. In contrast, MAQ+SAC and MAQ+RLPD required only 1 GPU hour each. Regarding the vanilla RL, IQL, and RLPD, the training required approximately 8 GPU hours per run, and SAC training required 1 GPU hour per run. This difference stems from the experimental setup: MAQ+IQL was trained in a single-environment setting, while MAQ+SAC and MAQ+RLPD were trained using 8 parallel environments. As a result, the training time for MAQ+IQL was approximately 8 times longer, and for MAQ+SAC and MAQ+RLPD, updates were performed 8 times per iteration.

**Hyperparameters**    All hyperparameters used in the online RL methods are detailed in Table 2, and those for Conditional VQVAE are listed in Table 3. For SAC, we used the default hyperparameters from Stable-Baselines3 [33]. For IQL [29], we adopted the PyTorch implementation by gwthomas's repository released on Github [34], which most closely follows the original paper. For RLPD [6], we used the official implementation released on GitHub.

In our MAQ-based methods, we made the following modifications. In MAQ+IQL, we built upon the IQL codebase and integrated the VQVAE decoder after the policy, enabling the policy to make decisions conditioned on the codebook size $K$. In MAQ+SAC, we extended the implementation of Discrete SAC [31] by incorporating the VQVAE decoder. For MAQ+RLPD, we built on MAQ+SAC and additionally implemented the symmetric sampling scheme from RLPD, using a symmetric ratio of 0.5 (equivalent to the offline ratio shown in Table 2).

**Dataset Segmentation**    For data partitioning, we used the human dataset provided by D4RL. Each task consists of 25 successful human demonstrations, which we split into training and testing sets in a 9:1 ratio, resulting in 22 trajectories for training and 3 for testing. We used the testing dataset to evaluate similarity between agents. The normalization procedure mentioned in the paper involves generating a random agent and using its performance on the testing dataset to normalize the similarity scores. Table 4 presents all the detailed results, including the raw data, training data, and testing data.

**Training Curves for All Environments**    Figure 6 presents the training curves of the experiments described in Subsection 5.2. The results are based on the normalized rewards across different environments. For Offline-to-Online RL (O2ORL) methods such as IQL and MAQ+IQL, training curves are plotted in environment steps and the x-axis is scaled by 0.5, so that 2M steps are compressed to the same span as 1M steps in other settings. Shaded regions indicate the standard deviation across three different random seeds.

**Environment Reward and Success Rate**    In D4RL, the success rate indicates whether the agent completes the task within the episode limit (e.g., opening the Door within 200 steps). In contrast, the normalized reward is a heuristic score defined by the environment that reflects how close the agent is to completing the task. For example, in the *Door* task, the reward increases as the door approaches a fully open state, but the episode is marked as successful only when the door is completely opened.

As shown in Figure 6 and summarized in Table 4, RLPD and MAQ+RLPD achieve similar success rates. However, their normalized rewards differ substantially: RLPD tends to maximize reward by opening the door quickly, whereas MAQ+RLPD opens the door more slowly and naturally while maintaining the same success rate.

**Computational Cost Analysis of MAQ**    Since there are 25 trajectories per task, the VQVAE can be trained in a very short time. Specifically, training VQVAE takes only 3 to 4 minutes, whereas RL

training requires several hours. While MAQ-based RL introduces some additional computational overhead, Table 5 presents a comparison of inference time for each RL algorithm, with and without MAQ.

It is also worth noting that MAQ can offer better inference efficiency, as it executes an entire macro action (e.g., 9 consecutive actions in the paper) with a single forward pass, whereas vanilla RL must do a forward pass at every timestep. In this case, MAQ-based RL can even achieve lower overall inference time than vanilla RL, especially when longer action sequences are used.

Table 2: Hyperparameters for training Adroit tasks.

| Parameter | BC | IQL | MAQ+IQL | SAC | MAQ+SAC | RLPD | MAQ+RLPD |
|---|---|---|---|---|---|---|---|
| Batch size | 256 | 256 | 256 | 128 | 128 | 256 | 128 |
| Learning rate | 3e-4 | 3e-4 | 3e-4 | 3e-4 | 3e-4 | - | - |
| Actor learning rate | - | - | - | - | - | 3e-4 | 3e-4 |
| Critic learning rate | - | - | - | - | - | 3e-4 | 1e-3 |
| Temperature learning rate | - | - | - | - | - | 3e-4 | 3e-4 |
| Optimizer | Adam | Adam | Adam | Adam | Adam | Adam | Adam |
| Offline training steps | 1M | 1M | 1M | - | - | - | - |
| Online training steps | - | 1M | 1M | 1M | 1M | 1M | 1M |
| Discount factor $\gamma$ | - | 0.99 | 0.99 | 0.99 | 0.99 | 0.99 | 0.99 |
| Warm-up steps | - | - | - | 1e2 | 8e3 | 1e4 | 8e3 |
| Update epoch | - | - | - | 1 | 8 | 1 | 8 |
| Value coeff | - | - | - | - | - | - | - |
| Entropy coeff | - | - | - | auto | auto | auto | auto |
| Offline ratio | - | - | - | - | - | 0.5 | 0.5 |
| Temperature alpha $\alpha$ | - | - | - | - | - | 0.2 | 1.0 |
| Replay buffer size | - | 2M | 2M | 1M | 1M | 1M | 1M |
| Target network update rate $\tau$ | - | 0.005 | 0.005 | 0.005 | 0.005 | 0.005 | 0.005 |
| Advantage coeff $\lambda$ | - | - | - | - | - | - | - |
| Asymmetric loss coeff $\tau$ | - | 0.7 | 0.7 | - | - | - | - |
| Inverse temperature $\beta$ | - | 3.0 | 3.0 | - | - | - | - |
| Std of Gaussian exploration noise | - | 0.03 | 0.03 | - | - | - | - |
| Range to clip noise | - | 0.5 | 0.5 | - | - | - | - |
| MAQ Codebook size $K$ | - | - | 16 | - | 8 | - | 16 |
| MAQ Macro action length $H$ | - | - | 9 | - | 8 | - | 9 |

Table 3: Hyperparameters of training MAQ's Conditional-VQVAE.

| Parameter | Adroit |
|---|---|
| Latent size | 256 |
| Learning rate | 3e-4 |
| Codebook size $K$ | [8, 16, 32] |
| Batch size | 32 |
| Commitment loss coeff $\beta$ | 0.25 |
| Macro length $H$ | [1 ... 9] |
| Optimizer | Adam |

Table 4: Detailed numerical results corresponding to the main table, including raw scores, training and testing data, and normalization baselines used for similarity evaluation.

| Tasks | | BC | IQL | MAQ+IQL | SAC | MAQ+SAC | RLPD | MAQ+RLPD | Training Dataset | Testing Dataset | Random |
|---|---|---|---|---|---|---|---|---|---|---|---|
| Door | DTW$_s$(↓) | 564.738 ± 39.892 | 451.199 ± 27.652 | **266.994 ± 26.499** | 820.125 ± 47.292 | **283.977 ± 34.359** | 672.954 ± 19.864 | **302.186 ± 19.757** | 285.085 ± 21.432 | 193.165 ± 30.694 | 643.789 ± 23.380 |
| | DTW$_n$(↓) | 700.093 ± 110.944 | 536.414 ± 33.763 | **237.304 ± 6.737** | 1285.045 ± 37.101 | **274.658 ± 24.079** | 823.852 ± 70.334 | **275.200 ± 41.988** | 299.011 ± 11.922 | 192.035 ± 12.539 | 1068.101 ± 26.872 |
| | WD$_s$(↓) | 6.721 ± 0.261 | 5.864 ± 0.263 | **4.388 ± 0.258** | 9.988 ± 0.127 | **4.561 ± 0.455** | 9.230 ± 0.213 | **4.587 ± 0.189** | 4.307 ± 0.134 | 3.013 ± 0.322 | 8.446 ± 0.188 |
| | WD$_n$(↓) | 6.017 ± 0.531 | 5.429 ± 0.109 | **3.446 ± 0.212** | 9.697 ± 0.162 | **3.701 ± 0.437** | 8.076 ± 0.116 | **3.734 ± 0.201** | 3.304 ± 0.118 | 2.176 ± 0.046 | 8.738 ± 0.093 |
| | Success(↑) | 0.020 ± 0.010 | 0.163 ± 0.055 | **0.930 ± 0.040** | 0.433 ± 0.232 | **0.563 ± 0.497** | 0.957 ± 0.067 | 0.933 ± 0.049 | 1.000 ± 0.000 | 1.000 ± 0.000 | 0.000 ± 0.000 |
| Hammer | DTW$_s$(↓) | 894.540 ± 24.821 | 887.642 ± 129.362 | **595.441 ± 63.226** | 1246.982 ± 131.710 | **607.021 ± 79.326** | 845.877 ± 52.838 | **578.816 ± 62.884** | 642.887 ± 48.775 | 459.715 ± 96.337 | 834.893 ± 45.341 |
| | DTW$_n$(↓) | 1056.832 ± 19.802 | 1083.427 ± 223.472 | **551.378 ± 72.545** | 1959.679 ± 118.135 | **570.395 ± 111.008** | 1178.736 ± 91.087 | **528.520 ± 81.925** | 654.830 ± 59.219 | 465.388 ± 104.728 | 1588.783 ± 81.516 |
| | WD$_s$(↓) | 8.792 ± 0.335 | 8.718 ± 0.696 | **5.243 ± 0.176** | 11.836 ± 0.509 | **5.893 ± 0.649** | 9.571 ± 0.443 | **5.202 ± 0.207** | 5.457 ± 0.166 | 3.892 ± 0.292 | 9.393 ± 0.251 |
| | WD$_n$(↓) | 6.670 ± 0.206 | 6.667 ± 0.586 | **3.371 ± 0.093** | 9.601 ± 0.234 | **3.751 ± 0.686** | 7.252 ± 0.185 | **3.320 ± 0.209** | 3.390 ± 0.077 | 2.394 ± 0.254 | 8.469 ± 0.105 |
| | Success(↑) | 0.000 ± 0.000 | **0.010 ± 0.010** | 0.000 ± 0.000 | 0.007 ± 0.012 | 0.000 ± 0.000 | **1.000 ± 0.000** | 0.557 ± 0.368 | 1.000 ± 0.000 | 1.000 ± 0.000 | 0.000 ± 0.000 |
| Pen | DTW$_s$(↓) | 745.007 ± 52.333 | 819.542 ± 37.199 | **737.831 ± 70.104** | 932.027 ± 80.278 | **726.706 ± 68.144** | 767.026 ± 94.645 | **740.454 ± 73.628** | 765.896 ± 61.689 | 556.756 ± 104.488 | 957.856 ± 64.804 |
| | DTW$_n$(↓) | 666.036 ± 16.540 | 688.794 ± 14.848 | **666.253 ± 29.119** | 950.579 ± 50.449 | **665.579 ± 34.040** | 722.633 ± 53.414 | **664.746 ± 38.493** | 727.791 ± 27.489 | 537.667 ± 62.392 | 845.175 ± 10.401 |
| | WD$_s$(↓) | 8.555 ± 0.782 | 8.865 ± 0.689 | **8.559 ± 0.793** | 10.498 ± 0.616 | **8.461 ± 0.737** | 9.211 ± 0.491 | **8.550 ± 0.736** | 8.112 ± 0.360 | 5.965 ± 0.285 | 12.305 ± 1.054 |
| | WD$_n$(↓) | 6.066 ± 0.695 | 6.166 ± 0.568 | **6.040 ± 0.727** | 8.212 ± 0.715 | **5.970 ± 0.687** | 7.101 ± 0.572 | **6.051 ± 0.672** | 5.868 ± 0.298 | 4.367 ± 0.336 | 9.275 ± 1.150 |
| | Success(↑) | 0.397 ± 0.025 | 0.400 ± 0.053 | **0.423 ± 0.065** | 0.320 ± 0.085 | 0.407 ± 0.006 | **0.617 ± 0.085** | 0.417 ± 0.045 | 1.000 ± 0.000 | 1.000 ± 0.000 | 0.000 ± 0.000 |
| Relocate | DTW$_s$(↓) | 654.866 ± 70.151 | 599.541 ± 102.262 | **443.821 ± 31.627** | 978.131 ± 102.574 | **579.314 ± 93.509** | 686.462 ± 65.869 | **564.726 ± 69.751** | 348.860 ± 25.274 | 201.109 ± 51.059 | 702.083 ± 91.180 |
| | DTW$_n$(↓) | 999.567 ± 214.562 | 931.336 ± 152.154 | **503.071 ± 18.131** | 1786.058 ± 216.580 | **731.311 ± 128.301** | 1205.731 ± 182.452 | **691.590 ± 132.459** | 410.301 ± 29.469 | 257.894 ± 49.510 | 1646.299 ± 229.892 |
| | WD$_s$(↓) | 8.812 ± 1.036 | 8.158 ± 0.445 | **7.389 ± 0.483** | 12.312 ± 0.445 | **7.903 ± 0.467** | 10.574 ± 0.525 | **8.048 ± 0.640** | 5.960 ± 0.360 | 3.627 ± 0.150 | 10.730 ± 0.308 |
| | WD$_n$(↓) | 6.937 ± 0.938 | 6.574 ± 0.319 | **5.345 ± 0.274** | 10.897 ± 0.254 | **5.692 ± 0.245** | 8.930 ± 0.237 | **6.127 ± 0.615** | 4.113 ± 0.212 | 2.559 ± 0.047 | 10.516 ± 0.329 |
| | Success(↑) | 0.013 ± 0.015 | 0.000 ± 0.000 | **0.203 ± 0.095** | 0.000 ± 0.000 | **0.143 ± 0.065** | 0.137 ± 0.025 | **0.173 ± 0.098** | 1.000 ± 0.000 | 1.000 ± 0.000 | 0.000 ± 0.000 |

Table 5: Comparison of computational costs with and without MAQ

|  | Vanilla RL (ms) | MAQ (ms) | Ratio (MAQ/Vanilla RL) |
|---|---|---|---|
| SAC/MAQ+SAC | 1.31 | 1.35 | 1.03 |
| IQL/MAQ+IQL | 0.81 | 1.12 | 1.39 |
| RLPD/MAQ+RLPD | 0.28 | 0.81 | 2.93 |

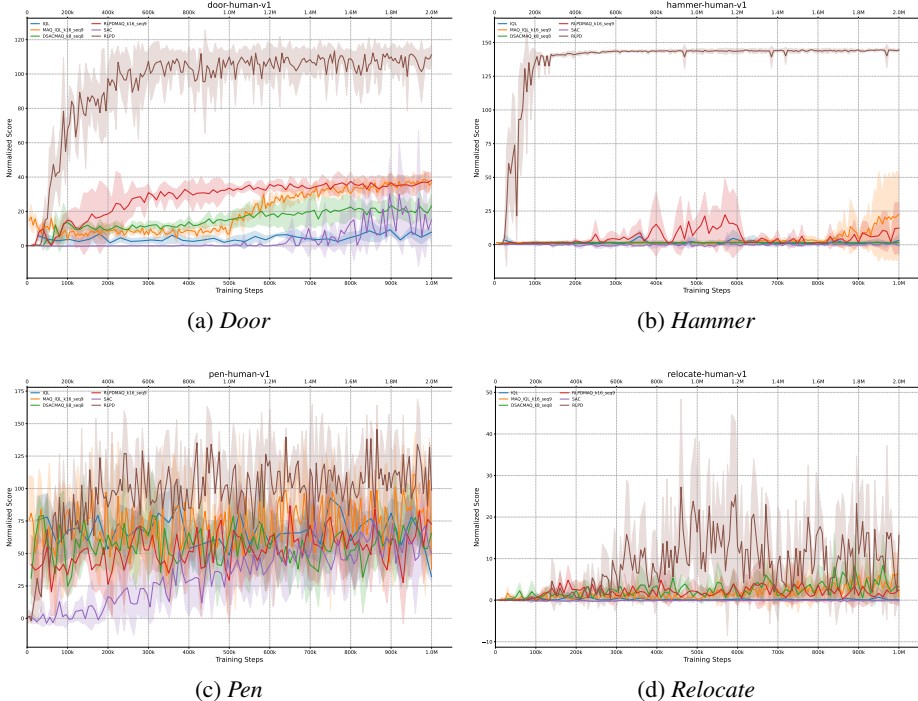

(a) *Door*

(b) *Hammer*

(c) *Pen*

(d) *Relocate*

Figure 6: Training curves for Adroit tasks.

# B   Further Analysis

In this section, we analyze MAQ from several aspects: (i) the effect of codebook size in Conditional-VQVAE (Appendix B.1); (ii) robustness to suboptimal demonstrations, including randomly shuffled and sorted datasets (Appendix B.2); (iii) the impact of extending the environment horizon on success rates in the *Hammer* task (Appendix B.3).

## B.1   Impact of Codebook Size on Similarity Score in MAQ

As discussed in Section 5.2.3, we previously observed that longer sequence lengths in VQVAE tend to lead to higher similarity scores when compared against human demonstrations. In this subsection, we further investigate whether varying the codebook size $K$ exhibits a similar trend. We detail the models used in this experiment and analyze how different codebook sizes influence the similarity scores across MAQ variants, including MAQ+IQL, MAQ+SAC, and MAQ+RLPD. It is important to note that the similarity score is a metric we define to approximate the behavioral similarity between agents and human demonstrations. While it provides a quantitative means of comparison, it does not capture the full semantics or intent of human-like behavior. As also mentioned in Section 5.2.3, in D4RL control tasks, the action space only includes the positions of the Shadow Hand, while the state space contains additional information about the target object. Accordingly, we evaluate how different codebook sizes affect the similarity score for different MAQ agents. Due to the computational constraints, we limited our experiments to VQVAEs trained with codebook sizes of 8, 16, and 32.

As shown in Figures 7, 8, and 9, we observe a consistent trend across all MAQ variants: as the macro action length increases, the similarity score also improves, regardless of the codebook size. Moreover, the similarity scores remain within a narrow variance range, indicating stable performance across different configurations.

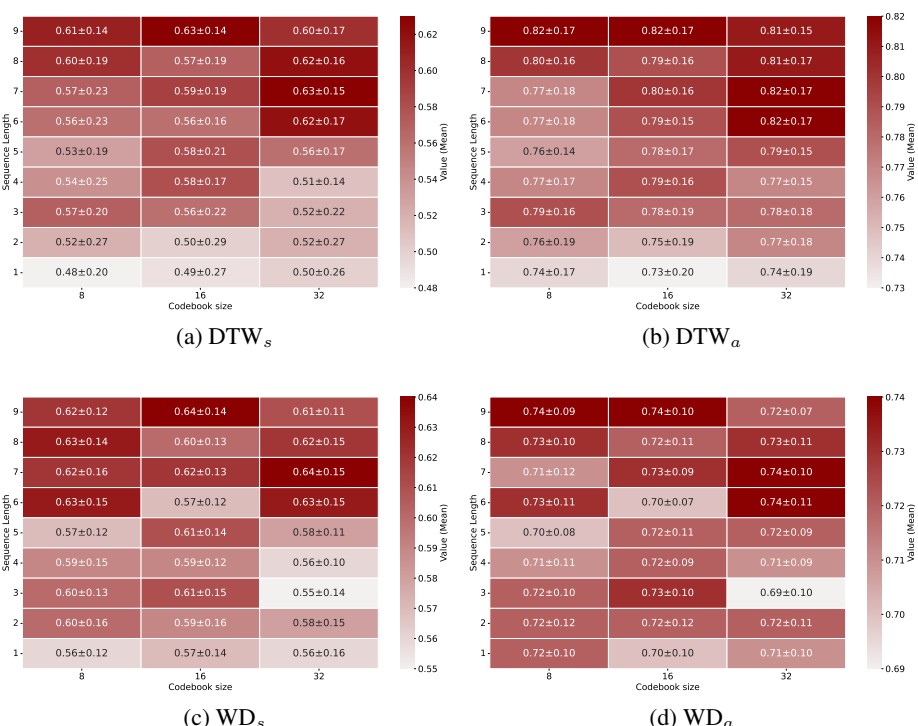

Figure 7: Similarity heatmaps of MAQ+IQL under different codebook sizes, measured by DTW and WD on state and action.

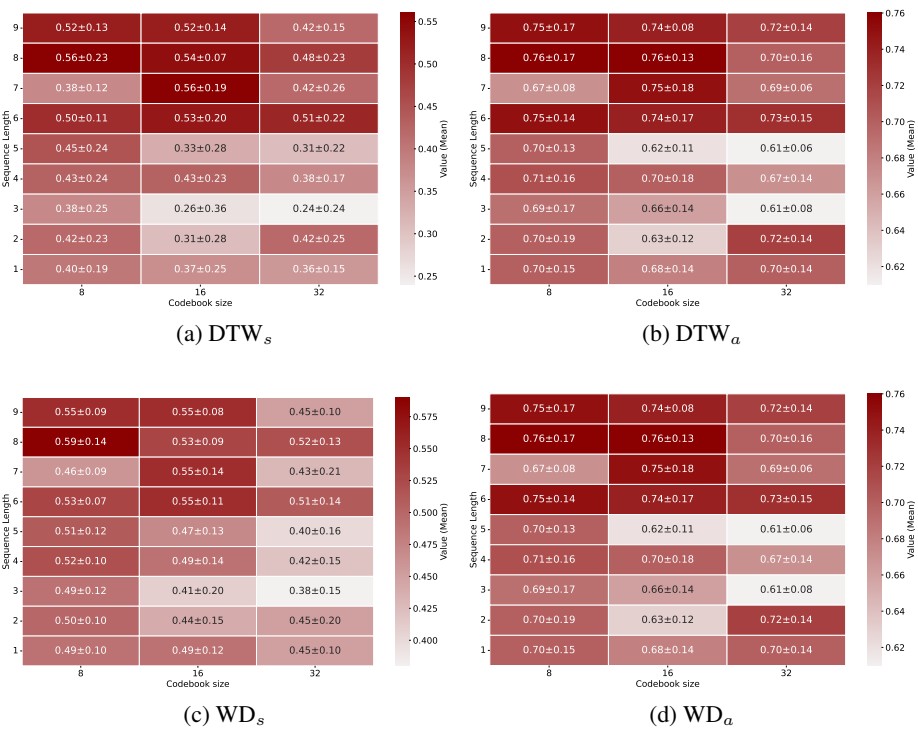

Figure 8: Similarity heatmaps of MAQ+SAC under different codebook sizes, measured by DTW and WD on state and action sequences.

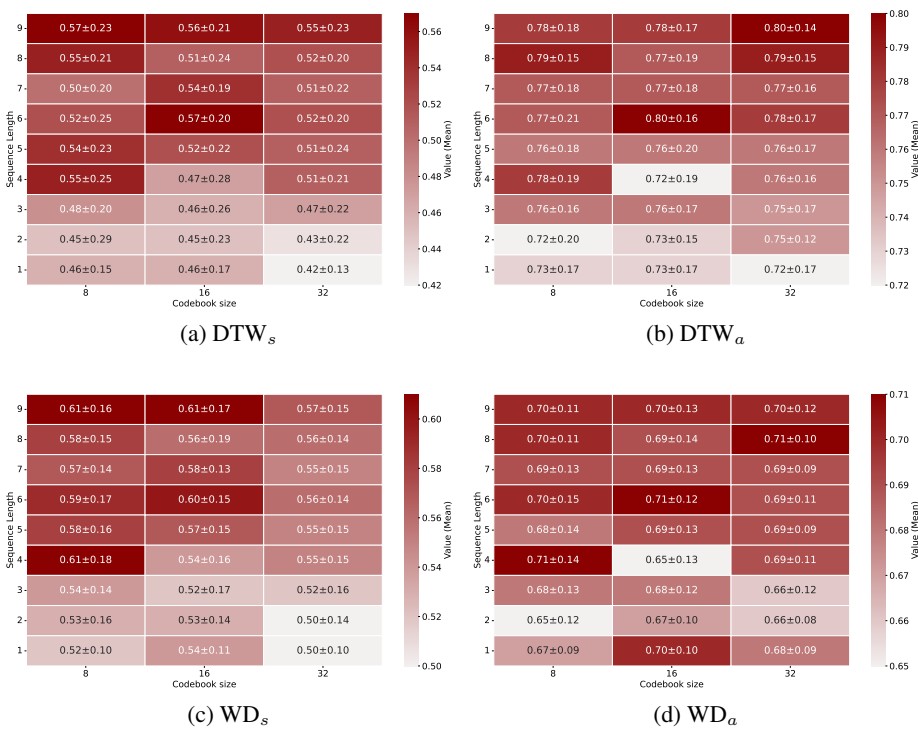

Figure 9: Similarity heatmaps of MAQ+RLPD under different codebook sizes, measured by DTW and WD on state and action.

In contrast, Figure 10 highlights how the effect of varying codebook sizes interacts differently with each underlying RL algorithm. For MAQ+IQL, which benefits from offline pretraining, success rates remain relatively consistent across different codebook sizes. MAQ+SAC, on the other hand, struggles under sparse reward conditions and fails to achieve a success rate above 30% regardless of codebook size or macro action length. Interestingly, MAQ+RLPD, which incorporates symmetric sampling, achieves a significantly higher success rate of up to 52%, demonstrating its advantage in such challenging environments.

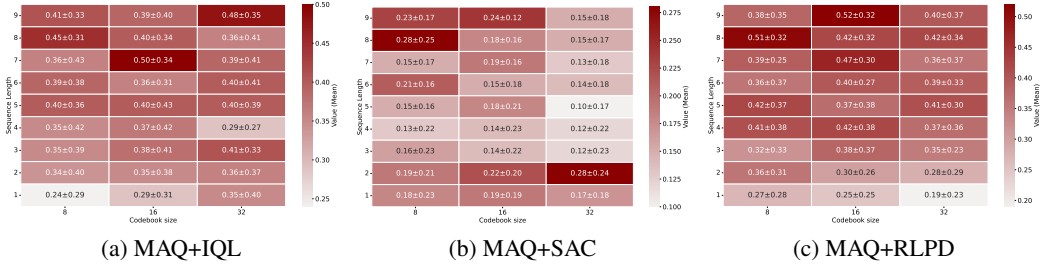

(a) MAQ+IQL          (b) MAQ+SAC          (c) MAQ+RLPD

Figure 10: Similarity heatmaps of MAQ agents under different codebook sizes, measured by success rates.

## B.2 Impact of Using Suboptimal Demonstration

In this subsection, we evaluate MAQ under different demonstration qualities, including suboptimal ones. We conduct an ablation study using subsets of trajectories with the lowest task rewards as well as random subsets. The results are shown in Table 6.

In this experiment, using the Pen task as an example, the success rate remains stable across demonstrations of varying quality. Moreover, MAQ trained on small subsets of demonstrations (e.g., 25% of the dataset under random sampling or the bottom 25% in terms of trajectory-wise reward) can still achieve higher similarity scores (DTW and WD values in the table are not normalized; lower is better). Even when using different random subsets, which may include higher-return trajectories, MAQ maintains a competitive success rate.

Table 6: Ablation of MAQ trained on suboptimal demonstrations. Similarity scores remain stable across demonstration ratios in **random** and **lowest**, and MAQ+RLPD achieves the best success rate and similarity.

| Tasks | | MAQ+RLPD | 75% random | 75% lowest | 50% random | 50% lowest | 25% random | 25% lowest |
|---|---|---|---|---|---|---|---|---|
| Door | $DTW_s(\downarrow)$ | 302.19 | 261.29 | **239.77** | 277.42 | 298.26 | 350.23 | 414.27 |
| | $DTW_a(\downarrow)$ | 275.20 | 257.20 | **242.36** | 263.66 | 294.37 | 335.20 | 408.44 |
| | $WD_s(\downarrow)$ | 4.59 | **4.38** | 4.70 | 4.42 | 4.77 | 5.19 | 5.85 |
| | $WD_a(\downarrow)$ | 3.73 | 3.37 | 3.85 | **3.27** | 3.77 | 4.10 | 4.49 |
| | Success($\uparrow$) | 0.93 | **0.98** | 0.79 | 0.85 | 0.55 | 0.48 | 0.03 |
| Hammer | $DTW_s(\downarrow)$ | 578.82 | **593.07** | 736.09 | 605.79 | 643.57 | 623.78 | 706.92 |
| | $DTW_a(\downarrow)$ | 528.52 | 519.13 | 641.82 | **515.70** | 550.62 | 545.67 | 519.61 |
| | $WD_s(\downarrow)$ | 5.20 | 5.37 | 5.54 | **5.15** | 5.26 | 5.48 | 6.34 |
| | $WD_a(\downarrow)$ | 3.32 | 3.37 | 3.53 | **3.21** | 3.40 | 3.40 | 3.58 |
| | Success($\uparrow$) | **0.56** | 0.33 | 0.31 | 0.30 | 0.28 | 0.23 | 0.51 |
| Pen | $DTW_s(\downarrow)$ | 740.45 | 679.96 | **590.02** | 669.14 | 649.56 | 686.26 | 686.74 |
| | $DTW_a(\downarrow)$ | 664.75 | 629.19 | **560.92** | 624.25 | 607.43 | 654.41 | 652.16 |
| | $WD_s(\downarrow)$ | 8.55 | 8.02 | 9.70 | **7.79** | 8.47 | 8.24 | 8.61 |
| | $WD_a(\downarrow)$ | 6.05 | 5.95 | 8.42 | **5.53** | 6.01 | 5.85 | 6.24 |
| | Success($\uparrow$) | **0.42** | 0.33 | 0.32 | 0.33 | 0.23 | 0.23 | 0.20 |
| Relocate | $DTW_s(\downarrow)$ | **564.73** | 582.06 | 596.79 | 641.00 | 625.72 | 652.59 | 637.79 |
| | $DTW_a(\downarrow)$ | **691.59** | 739.38 | 741.85 | 857.08 | 772.72 | 862.77 | 853.56 |
| | $WD_s(\downarrow)$ | 8.05 | **7.84** | 8.08 | 8.34 | 8.83 | 8.87 | 8.57 |
| | $WD_a(\downarrow)$ | 6.13 | 6.11 | **5.89** | 6.54 | 6.47 | 6.60 | 6.32 |
| | Success($\uparrow$) | **0.17** | 0.08 | 0.04 | 0.04 | 0.02 | 0.00 | 0.01 |

## B.3 Impact of Different Environment Horizon

In Appendix A, we discussed the distinction between environment reward and success rate, as well as the score gap between RLPD and MAQ+RLPD in the training curves. In the *Hammer* task, the success rates of RLPD and MAQ+RLPD still differ markedly. We conducted a deeper analysis of this task. We found that in the 25 human demonstration trajectories, humans take an average of 451.4 steps to successfully complete the task. However, in the D4RL environment, the agent is restricted to a maximum of 200 steps per trajectory, after which the episode is terminated and directly considered a failure.

Since MAQ learns from human demonstrations, it tends to favor smooth but slow movements, while non-human-like RL agents tend to complete the task more quickly but less naturally. To validate this hypothesis, we increased the maximum episode length from 200 to 450 steps. As shown in Table 7, MAQ+RLPD's success rate improves significantly from 0.56 to 0.72 while maintaining human-likeness scores.

Table 7: Ablation of MAQ+RLPD on *Hammer* under different training horizons $h$.

| Tasks | | RLPD ($h = 200$) | MAQ+RLPD ($h = 200$) | MAQ+RLPD ($h = 450$) |
|---|---|---|---|---|
| Hammer | $DTW_s(\downarrow)$ | 845.88 | **578.82** | 648.32 |
| | $DTW_a(\downarrow)$ | 1178.74 | **528.52** | 637.25 |
| | $WD_s(\downarrow)$ | 9.57 | **5.20** | 6.17 |
| | $WD_a(\downarrow)$ | 7.25 | **3.32** | 4.15 |
| | Success($\uparrow$) | 1.00 | **0.56** | 0.72 |

# C   Details of Human-likeness Survey

## C.1   Survey Setup and Evaluation Protocol

This subsection describes the interface shown to the evaluators. Each questionnaire contains four "tasks," where each task is split into two stages, and each stage consists of a series of two-alternative forced-choice (2AFC) trials.

**Stage 1: Human-Detection Phase (Figure 11a).**   The assessors are told that *at least one* of the two clips was produced by a human and must identify which one. They also mark their confidence on a five-point scale, but our main analysis considers only the binary choice and ignores the confidence scores.

**Stage 2: Ranking Phase (Figure 11b).**   Here, we test whether the behavior generated by MAQ-based agents appears more human-like than that of baseline agents or even the human demonstrations. Model clips (optionally mixed with human reference clips) are presented in 2AFC pairs that are scheduled with a shuffled single-elimination and round-robin *mini-tournament*. Up to eight clips yield no more than 17 head-to-head comparisons; each win earns one point. Clips are ranked by win-rate $w_i := \frac{\text{wins}_i}{\text{appearances}_i}$, with mean reported confidence used only to break ties, producing an ordering from the least to the most human-like one.

**Post-ranking feedback.**   After the Ranking Phase, each evaluator is shown the clips judged the *most* and the *least* human-like and may optionally explain those judgments in free text (Figure 11c). We analyze these comments in Section C.2.

## C.2   Additional Results of Human Feedback on Behavior Analysis

In this subsection, we discuss the additional behavioral results and the qualitative feedback received for each game. Figure 12 displays the distribution of the Top-1 and Top-2 rankings obtained in the Ranking Phase. Across all games, human demonstrations achieve the highest Top-1 probability, confirming that evaluators can reliably identify genuine human behavior. Among the learned agents, MAQ+IQL attains the second-highest Top-1 rate, followed by MAQ+RLPD, showing that our methods frequently convince evaluators that their behavior is indeed human-like.

In the *Hammer* task, five evaluators placed the human demonstration at Rank 1. The next most human-like agents were the MAQ variants—MAQ+RLPD and MAQ+IQL, each with four first-place votes, followed by MAQ+SAC with one first-place vote.

**What evaluators liked about MAQ policies.**   Comments converge on three strengths: deliberate grip preparation, a realistic multi-strike rhythm, and a smooth follow-through.

> "*Is able to hammer the nail.*"
> "*First takes hold of the hammer; because you need to aim, the first hit is lighter and the second harder.*"
> "*Humans can aim accurately when hammering a nail and probably won't drive the nail completely in at once.*"
> "*It performs smoothly and hits the nail several times.*"
> "*There is a back-and-forth hammer-swinging motion.*"
> "*The feeling of driving it in on the last strike is very human-like.*"

**Why baseline agents were judged less human-like.**   Non-MAQ policies drew noticeably harsher remarks:

> "*Can't even lift the hammer.*"
> "*It threw the hammer.*"
> "*It fails the task.*"
> "*It cannot even lift the hammer.*"
> "*Hits too far from the nail and releases the hammer in an unnatural manner.*"

MAQ does more than replicating the gross kinematics of hammering: several evaluators remarked that the policy "re-aims and strikes again" after an initial impact, a behavior they naturally expect from humans. Figure 13 contrasts MAQ+RLPD with the plain RLPD agent. MAQ+RLPD delivers a series of well-timed blows that gradually seat the nail, whereas RLPD drives the nail in a single, overly forceful hit. Although MAQ+RLPD does not chase the maximum task score, it acts purposefully the way a person would, delivering several well-timed blows, while vanilla RLPD, trained only to maximize reward, drives the nail in a single, overly forceful hit that evaluators consistently judged "unlike a human."

In the *Pen* task, six evaluators placed **MAQ+IQL** at Rank 1 more than any other policy, while vanilla RLPD received five first-place votes and the human demonstration received only two.

**What evaluators liked about MAQ policies.**   Written feedback centers on a natural, well-coordinated grip and fine adjustments that keep the pen aligned with the target:

> "*It is using all the fingers.*"
> "*It feels like all fingers are used.*"
> "*Humans would likely adjust the pen to be as close as possible to the target model.*"
> "*The middle and ring fingers move with the rest of the hand, which aligns with ergonomic principles.*"
> "*Holding the pen like this is very steady and human-like.*"
> "*The finger angles are not too strange and the task is completed quickly.*"

**Why baseline agents were judged less human-like.**   Agents trained without MAQ conditioning were criticized for awkward finger placement and lack of re-aiming:

> "*It's unnatural for a human to leave one finger open when trying to grasp.*"
> "*I feel like fingers are out of control.*"
> "*After holding the pen, I wouldn't deliberately stick out a single finger.*"
> "*It does not adjust the pen.*"
> "*He basically shows no intention of aiming or aligning.*"
> "*People don't hold a pen vertically.*"
> "*The angles of the fingers appear twisted.*"

These remarks echo the ranking statistics: MAQ conditioning encourages full-hand coordination and incremental alignment, which are features that evaluators intuitively associate with human pen manipulation, whereas reward-only policies often adopt grasp patterns that look distinctly non-human.

In the *Relocate* task, six evaluators placed the human demonstration at Rank 1, but **MAQ+IQL** was a close second with five first-place votes, followed by MAQ+RLPD (three) and MAQ+SAC (one).

**What evaluators liked about MAQ policies.**   Positive remarks emphasize three recurring traits: a direct, purposeful approach to the ball, continuous motion once the object is secured, and smooth point-to-point transfer.

> "*It moves towards the ball.*"
> "*After the ball is grasped, the motion continues without any pause.*"
> "*Humans should smoothly move an object from the source to the destination point the hand is gently and smoothly picking, moving, and putting.*"
> "*At least it puts the ball near the target position.*"
> "*The hand shows slight grasping motions, and the movement trajectory feels quite natural.*"
> "*People normally go straight to grab the ball.*"

**Why baseline agents were judged less human-like.**   When an agent broke the direct-and-smooth pattern, the evaluators reacted strongly:

*"It moves away from the ball."*
*"The hand stopped moving before it actually grasped the ball."*
*"It looks like it can't even pick up the ball."*
*"It does not move the object at all."*
*"The arm and finger movements are both very chaotic."*
*"When I grab the ball, I would align my* palm *to the ball, not my wrist."*

Taken together, these comments show that MAQ conditioning steers agents toward the straight-line reach, uninterrupted grasp-and-place sequence that evaluators intuitively associate with human relocation behavior, whereas reward-only policies often hesitate, wander, or execute awkward wrist-first contacts that look distinctly non-human.

**Summary of MAQ Advantages.**    Across all appendix tasks, including *Hammer*, *Pen*, and *Relocate*, the MAQ-conditioned agents consistently receive the highest human-likeness rankings among the learned policies. Evaluators repeatedly highlight three qualities that MAQ alone imparts:

1. **Preparatory alignment.** MAQ policies re-aim, re-grip, or re-strike in ways that mirror how humans make minor adjustments before the decisive action.

2. **Appropriate force modulation.** MAQ agents hammer with several well-timed taps, guide the pen with gentle finger coordination, and transport the ball with a continuous grasp-and-place sequence—behaviors evaluators intuitively regard as natural.

These observations reinforce our central claim: MAQ does not simply optimize task scores; it systematically shapes behavior toward the timing, grip strategy, and motion fluidity that humans recognize as their own.

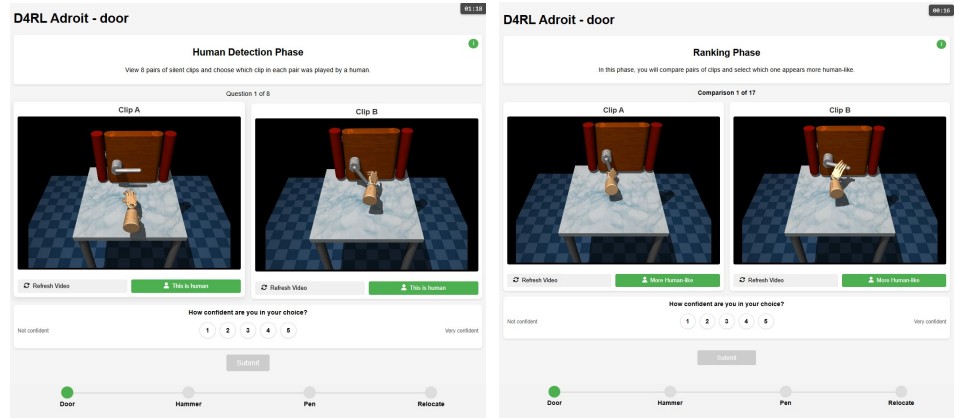

(a) Human Detection Phase.        (b) Ranking Phase.

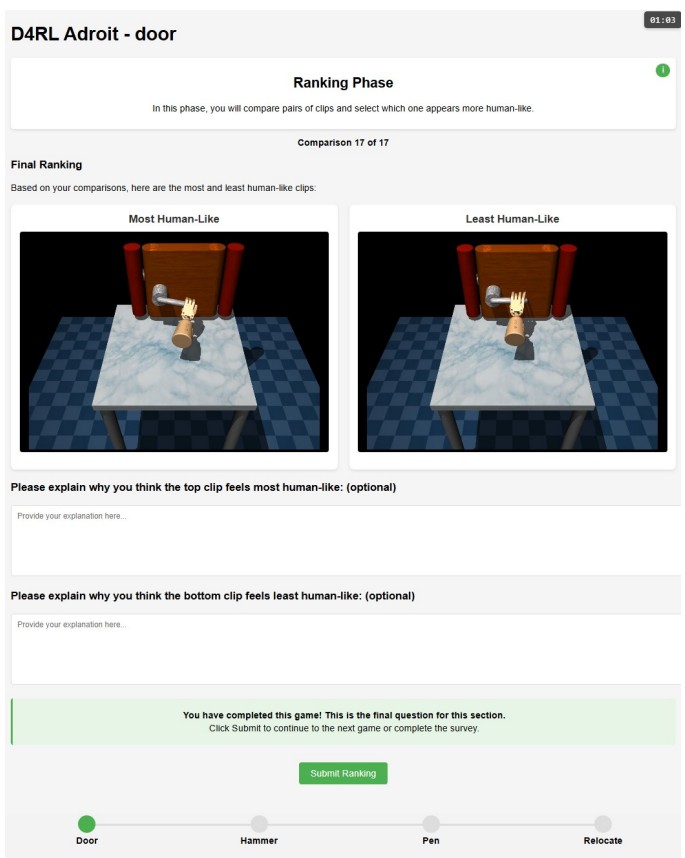

(c) Post-ranking feedback.

Figure 11: Evaluation protocol.

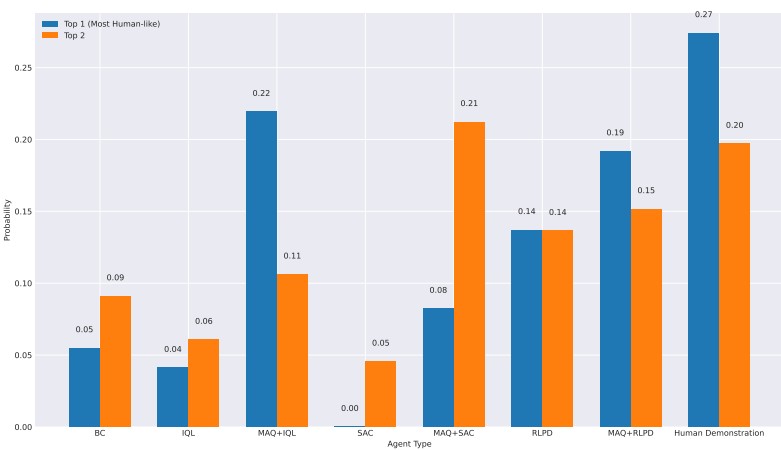

Figure 12: Probability distribution of agents ranked top 1 and top 2 in Ranking Phase.

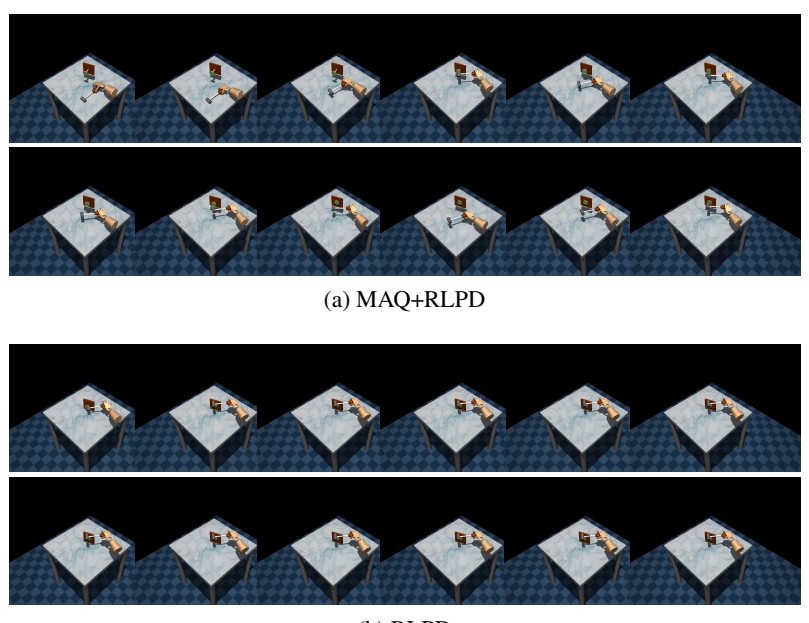

(a) MAQ+RLPD

(b) RLPD

Figure 13: Agent behavior in *Hammer*.

