# OpenReview forum: "Learning Human-Like RL Agents Through Trajectory Optimization With Action Quantization"
_NeurIPS.cc/2025/Conference — NeurIPS 2025 poster_

### Official Review · Reviewer_gwPt · 2025-06-30

**Clarity:** 3
**Significance:** 3
**Originality:** 2
**Rating:** 4
**Confidence:** 4

**Summary:**

The paper addresses the challenge of creating RL agents that not only excel at tasks but also behave in a manner that resembles human actions.
To tackle this, the paper formulates human-likeness as a trajectory optimization with human constraints within the RHC framework.
Based on the formulation, the paper proposes a method (MAQ) to distill human demonstrations into discrete macro actions using VQVAE and training the RL agent in this macro action space, biasing it toward more human-like behavior.
Empirical evaluation of the MAQ-enhanced agents show improved alignment with human demonstration trajectories (both in metrics and human evaluations) without sacrificing task performance.

**Questions:**

- Human evaluation settings: The exact number of evaluators for the human study and the agreement among evaluators are not provided to ensure statistical robustness.

- Additional computational costs: Training the VQVAE component can lead to more computational costs and training time. A cost comparison between the MAQ-based RL training and vanilla RL training should be conducted and discussed.

**Ethical Concerns:**

["NO or VERY MINOR ethics concerns only"]

**Final Justification:**

The authors conduct detailed responses, including additional benchmark and baselines, explanation on Negative impact on task success, ablation studies, etc. If these responses can be integrated into the revised paper, the quality will be improved a lot.

**Limitations:**

yes

**Quality:**

2

**Strengths And Weaknesses:**

Strengths:

- Important problem: Addressing the gap between high-performing RL agents and human-like behavior is important, especially for applications where trust, interpretability, and natural behavior are essential (e.g., human–robot interaction and social agents).

- Human evaluation included: The experimental evaluations provide both quatitative (trajectory similarity metrics) and qualitive results (Turing Test and human-likeness ranking) to demonstrate the efficiency of incorporating MAQ (distilled from human demonstrations) into RL training to improve human likeness.

Weaknesses:

- Limited task/domain evaluation: The experiments are confined to a limited benchmark set (Adroit tasks). While this can be a strong testbed, the generality beyond this, especially those tasks or domains heavily demanding for human-like behaviors (e.g., navigation, games, role-playing), is not empirically validated.

- Negative impact on task success (RLPD): The results for MAQ+RLPD suggest MAQ can sometimes negatively impact task performance, especially when the underlying RL agent is highly performant. This needs a more nuanced discussion or a proposed mitigation.

- Insufficient ablation studies: Lack experimental analysis on the (least) amount of human demonstrations required for MAQ-based RL agents, and the the codebook size K in VQVAE.

- Lack of strong IL baselines: Comparison only to BC and not to state-of-the-art imitation learning or inverse RL baselines.

- Limited novelty:  While the particular integration with RHC and the specific quantization method represent an incremental but meaningful improvement, the combination of IL and macro-action frameworks is not entirely new. So the novelty is somewhat restricted by the reliance on existing architectures and frameworks (e.g., VQVAE and RHC), even though the way they are combined is original.

---

> ### Author Rebuttal · Authors · 2025-07-31
>
> We thank the reviewer for constructive feedback and thoughtful reviews. We address each concern and question as follows.
>
> > W1: Limited task/domain evaluation: The experiments are confined to a limited benchmark set (Adroit tasks). While this can be a strong testbed, the generality beyond this, especially those tasks or domains heavily demanding for human-like behaviors (e.g., navigation, games, role-playing), is not empirically validated.
>
> Yes, as the reviewer noted, we chose Adroit because it is a strong testbed, which requires continuous control and fine-grained skills. We believe it is well-suited for evaluating human-like manipulation behavior.
>
> However, we also agree with the reviewer that demonstrating the generality of MAQ is important. Therefore, we conduct additional experiments in two Atari environments:
> * Venture: a navigation and role-playing style task in which player movements are clearly visible
> * Enduro: a third-person racing game with a highly dynamic environment
>
> Next, we train both PPO and MAQ+PPO on these games. For MAQ, the macro-action sequence length is set to 9. To assess human-likeness in the discrete action space, we adopt the method proposed in [1], which calculates the Wasserstein distance between action distributions, where xN represents the aggregated action histogram over N steps. The results below show that MAQ+PPO outperforms PPO in both scores and human-likeness, demonstrating that MAQ can generalize beyond Adroit tasks. We would be happy to include these additional experiments in the revision if the reviewer finds them satisfactory.
>
> |||PPO|MAQ+PPO|
> |-|-|-|-|
> |Enduro|WD_a x1|0.86|**0.45**|
> ||WD_a x8|2.49|**1.55**|
> ||WD_a x16|3.55|**2.59**|
> ||Score|25.66|**425.33**|
> |||||
> |Venture|WD_a x1|1.19|**0.23**|
> ||WD_a x8|3.26|**0.91**|
> ||WD_a x16|1.67|**1.56**|
> ||Score|66.66|**436.66**|
>
> \* Note: DTW and WD are not normalized; lower is better.
>
> [1] Pearce, Tim, et al. "Imitating human behaviour with diffusion models." arXiv preprint arXiv:2301.10677 (2023).
>
> > W2: Negative impact on task success (RLPD): The results for MAQ+RLPD suggest MAQ can sometimes negatively impact task performance, especially when the underlying RL agent is highly performant. This needs a more nuanced discussion or a proposed mitigation.
>
> This is a great question! We were also curious about why MAQ+RLPD performed worse than RLPD, especially in the Hammer environment, where the success rate drops from 1.00 to 0.56. After submitting the paper, we conducted a deeper analysis of this task. We find that in the 25 human demonstration trajectories, humans take an average of 451.4 steps to successfully complete the task. However, in the D4RL environment, the agent is restricted to a maximum of 200 steps per trajectory, after which the episode is automatically considered a failure.
>
> Since MAQ learns from human demonstrations, it tends to favor smooth but slow movements, while non-human-like RL agents tend to complete the task more quickly but less naturally. To validate this hypothesis, we increased the maximum episode length from 200 to 450 steps. As shown in the table below, MAQ+RLPD's success rate improves significantly from 0.56 to 0.72 while maintaining human-likeness scores. We will include this analysis with an explanation in the revision. Thank you for highlighting this!
>
> ||RLPD|MAQ+RLPD (200)|MAQ+RLPD (450)|
> |-|-|-|-|
> |DTW$_s$ (↓)|845.88|578.82|648.32|
> |DTW$_a$ (↓)|1178.74|528.52|637.25|
> |WD$_s$ (↓)|9.57|5.20|6.17|
> |WD$_a$ (↓)|7.25|3.32|4.15|
> |Success|1.00|0.56|0.72|
>
> \* Note: DTW and WD are not normalized; lower is better.
>
> > W3: Insufficient ablation studies: Lack experimental analysis on the (least) amount of human demonstrations required for MAQ-based RL agents, and the the codebook size K in VQVAE.
>
> Regarding the amount of human demonstrations, we conduct an ablation study using different amounts of training data in Door and Pen environments: 90% (as in the paper), 75%, 50%, and 25%. The results below show that while the success rate decreases as the amount of training data is reduced (as expected), the trajectory similarity scores remain consistent. This demonstrates that MAQ can still capture human-like behavior even with as few as 6 trajectories.
>
> |||90% (22:3)|75% (18:7)|50% (12:13)|25% (6:19)|
> |-|-|-|-|-|-|
> |Door|DTW$_s$ (↓)|302.19|261.29|277.42|350.23|
> ||DTW$_a$ (↓)|275.20|257.20|263.66|335.20|
> ||WD$_s$ (↓)|4.59|4.38|4.42|5.19|
> ||WD$_a$ (↓)|3.73|3.37|3.27|4.10|
> ||Success|0.93|0.98|0.85|0.48|
> |||||||
> |Pen|DTW$_s$ (↓)|740.45|679.96|669.14|686.26|
> ||DTW$_a$ (↓)|664.75|629.19|624.25|654.41|
> ||WD$_s$ (↓)|8.55|8.02|7.79|8.24|
> ||WD$_a$ (↓)|6.05|5.95|5.53|5.85|
> ||Success|0.42|0.33|0.33|0.23|
>
> \* Note: DTW and WD are not normalized; lower is better.
>
> Regarding the codebook size K, please refer to our Appendix. C: "Impact of Codebook Size on Similarity Score in MAQ", which provides a detailed analysis of how different values of $K$ affect performance.
>
> > W4: Lack of strong IL baselines: Comparison only to BC and not to state-of-the-art imitation learning or inverse RL baselines.
>
> We include BC as a baseline to represent a simple and widely used method. As the reviewer noted, BC captures human-likeness to some degree but does not optimize with task success rate. We did not include GAIL for several reasons. First, GAIL relies on high-quality expert demonstrations, while our focus is on imitating human behavior, which can be suboptimal but diverse. Second, GAIL is known to require extensive training time to train in high-dimensional continuous control tasks. For example, training GAIL on the 6-dimensional Walker task takes ~25M steps. In contrast, our method learns macro actions for the 24-dimensional Pen task in only 1M steps. We also attempted to train GAIL for 1M steps on the four D4RL tasks, but GAIL failed to produce any successful trajectories, possibly requiring more training and computational resources.
>
> However, we agree that comparing other imitation learning methods would strengthen our results. To this end, we included IQ-Learn [1], one of the strongest IL algorithms. Since IQ-Learn was not originally tested on D4RL tasks, we applied it (using the hyperparameters recommended in the original paper) to all four tasks. The results below show that IQ-Learn performs poorly in both human-likeness and task success, while MAQ+IQ-Learn improves both performance. We believe this additional experiment offers a more convincing comparison and also demonstrates that MAQ is complementary to IL algorithms. We can include this experiment if the reviewer finds it satisfactory.
>
> [1] Garg, Divyansh, et al. "Iq-learn: Inverse soft-q learning for imitation." Advances in Neural Information
>
> |||IQ-Learn|MAQ+IQ-Learn|
> |-|-|-|-|
> |Door|DTW$_s$ (↓)|633.88|358.11|
> ||DTW$_a$ (↓)|796.35|345.33|
> ||WD$_s$ (↓)|9.10|5.11|
> ||WD$_a$ (↓)|8.18|4.04|
> ||Success|0.00|0.25|
> ||
> |Hammer|DTW$_s$ (↓)|844.00|627.05|
> ||DTW$_a$ (↓)|1261.88|607.84|
> ||WD$_s$ (↓)|9.58|6.25|
> ||WD$_a$ (↓)|7.41|4.18|
> ||Success|0.00|0.04|
> ||
> |Pen|DTW$_s$ (↓)|1025.75|778.53|
> ||DTW$_a$ (↓)|768.69|684.22|
> ||WD$_s$ (↓)|10.32|8.86|
> ||WD$_a$ (↓)|7.11|6.38|
> ||Success|0.09|0.20|
> ||
> |relocate|DTW$_s$ (↓)|641.08|534.19|
> ||DTW$_a$ (↓)|1261.29|591.09|
> ||WD$_s$ (↓)|10.20|7.72|
> ||WD$_a$ (↓)|8.77|5.62|
> ||Success|0.00|0.02|
>
> \* Note: DTW and WD are not normalized; lower is better.
>
> > W5: Limited novelty ...
>
> Our contribution lies in how we integrate these existing components to address the underexplored challenge of achieving human-like behavior in RL. We agree with the reviewer that VQVAE and RHC are not novel in themselves. However, to the best of our knowledge, this work is the first to combine macro actions learned from human demonstrations via VQVAE with RHC to achieve both high task performance and human-likeness across multiple RL algorithms.
>
> While previous human-like RL methods often rely on handcrafted behavioral cost functions, our approach offers a general framework that avoids manual design. We believe this represents a meaningful and sufficiently novel contribution, and a significant step forward for the human-like RL community.
>
> > Q1: Human evaluation settings …
>
> There are 19 human evaluators in our human study. We will include this information in the revision. Thank you for pointing it out. We will clarify this in the revision. In addition, we follow the similar method used in [2] reporting 96% confidence intervals for both the Turing Test and the ranking test. Thank you again for raising this point.
>
> [2] Tirinzoni, Andrea, et al. "Zero-shot whole-body humanoid control via behavioral foundation models." arXiv preprint arXiv:2504.11054 (2025).
>
> > Q2: Additional computational costs …
>
> Since we only have 25 trajectories per task, the VQVAE can be trained in a very short time. Specifically, training VQVAE takes only 3 to 4 minutes, whereas RL training requires several hours. While MAQ-based RL introduces some additional computational overhead, we conduct a comparison of inference time for each RL algorithm, with and without MAQ, as shown in the table below. Overall, the cost remains acceptable.
>
> ||Vanilla RL (ms)|MAQ (ms)|Ratio (MAQ/Vanilla RL)|
> |-|-|-|-|
> |SAC/MAQ+SAC|1.31|1.35|1.03|
> |IQL/MAQ+IQL|0.81|1.12|1.39|
> |RLPD/MAQ+RLPD|0.28|0.81|2.93|
>
> It is also worth noting that MAQ can offer faster inference efficiency, as it executes an entire macro action (e.g., 9 consecutive actions in the paper) with a single forward pass, whereas vanilla RL must forward at every timestep. In this case, MAQ-based RL can even achieve faster overall inference time than vanilla RL, especially when longer action sequences are used. We will include these results with a detailed explanation of the computational cost in the revision. Thank you for your suggestion.

---

> > ### Comment · Reviewer_gwPt · 2025-08-05
> > **Response to Authors**
> >
> > Thanks for your responses which have addressed my concerns. I would like to raise my score.

---

> ### Author Response · Authors · 2025-08-05
>
> We thank the reviewer for all the constructive feedback and the time the reviewer put into helping us improve our submission.

---

### Official Review · Reviewer_HJQk · 2025-06-30

**Clarity:** 3
**Significance:** 2
**Originality:** 3
**Rating:** 4
**Confidence:** 4

**Summary:**

This paper formulates the challenge of producing human-like behavior in reinforcement learning (RL) as a trajectory optimization problem. To address this, the authors introduce Macro Action Quantization (MAQ), a framework that distills human demonstrations into macro-actions using a Conditional Vector Quantized Variational Autoencoder (VQVAE). The VQVAE is trained to encode sub-trajectories of length H into a discrete codebook of size K. A policy is then learned over this discrete action space, selecting macro-actions via codebook indices to maximize environment rewards. Each selected index is decoded into a sequence of primitive actions conditioned on the current state.

Empirical evaluation on the D4RL Adroit benchmark shows that integrating MAQ with standard RL algorithms (IQL, SAC, RLPD) leads to a significant improvement in human-likeness, as measured by Dynamic Time Warping and Wasserstein distance, without substantial loss in task performance. Additionally, a human evaluation study is conducted using a Turing Test style setup and a pairwise ranking test, demonstrating that MAQ-trained agents are consistently judged to behave more like humans compared to baseline agents.

**Questions:**

1. Statistical Validity of Human Evaluation: How many human evaluators participated in the Turing Test and ranking tests? Please include this number and report significance tests to support the reliability of these results.
2. RLPD Degradation: In Table 1 and Appendix Figure 6, RLPD seems to suffer significantly when combined with MAQ. Can the authors clarify this discrepancy between reported success rate and observed reward curves? Can you provide more details on how the success rate is computed vs reward curve normalized score
3. Demonstration Quality: Were the human demonstrations used in the evaluation (e.g., Turing Test) all successful executions? Were agent demonstrations sampled from successful rollouts? This information is crucial for fair comparison.
4. Motivation and Impact: The paper briefly mentions the importance of human-likeness but does not provide a compelling argument or citations. Could the authors expand on the value and implications of designing human-like agents, including societal risks or limitations?

**Ethical Concerns:**

["NO or VERY MINOR ethics concerns only"]

**Final Justification:**

The rebuttal effectively clarified several concerns, e.g., RLPD degradation, evaluation details, and demonstration quality. However, motivation for human-likeness, broader impact discussion, and figure clarity remain for future edits. The method is technically solid but limited in significance, so I keep my overall rating, although quality has gone up.

**Limitations:**

The paper does not adequately discuss the limitations or potential societal impacts of building human-like agents. The Discussion section does not meaningfully reflect on possible negative consequences or ethical concerns (e.g., manipulation, human trust, or deceptive agents). I recommend the authors include a Broader Impacts section that honestly addresses these issues and situates their work in a wider context.

**Quality:**

3

**Strengths And Weaknesses:**

**Strengths:**

- The paper invests in analyzing the human-likeness of policies trained with RL, which, although tangentially implicit when using human data for training, is a relatively novel evaluation of the actual behavioral outcome. A Turing Test-style human evaluation provides additional validation of perceived human-likeness.
- The proposed MAQ framework integrates well with existing RL algorithms and empirically improves human-likeness metrics across tasks.

**Weaknesses:**

- Figures are poorly described, particularly Figure 1, which lacks subfigure explanations and adequate detail.
- If right, The evaluation uses only ~2.5 human demonstration trajectories per task, raising concerns about robustness of human likenliness mesures
- There doesn’t seem to be a clear observation  that MAQ  degrades task performance for the highest scoring RL algorithm, RLPD. Difference is even more obious when looking at the appendix reward curves, Figure 6.
- The motivation for human-like behavior is weakly argued and under-cited, beyond bringing in discussion the Turing Test.
- The number of human evaluators is not reported, and no statistical significance analysis is provided for the Turing Test or ranking results.
- The paper lacks a Broader Impact or Societal Impact discussion, especially relevant when aiming to mimic human behavior.
- It remains unclear whether demonstrations used in the human-likeness evaluation are successful ones. This could unfairly bias comparisons.

---

> ### Author Rebuttal · Authors · 2025-07-31
>
> We thank the reviewer for constructive feedback and thoughtful reviews. We address each concern and question as follows.
>
> > W1: Figures are poorly described, particularly Figure 1, which lacks subfigure explanations and adequate detail.
>
> Thank you for the suggestion. We will revise the caption to include clearer subfigure explanations and more detailed descriptions in the revision.
>
> > W2: If right, The evaluation uses only ~2.5 human demonstration trajectories per task, raising concerns about robustness of human likenliness mesures
>
> Yes, due to the limited number of human demonstration trajectories (25 per task), we adopted a 9:1 train/test to maximize the training data available for learning human behavior. As a result, the test data contains only 3 trajectories. To mitigate concerns about robustness, each algorithm in the paper is trained three times using three different randomly selected train/test splits (while maintaining the 9:1 ratio). This reduces the chance of overfitting to any particular test data and helps ensure the evaluation results are reliable and robust.
>
> In addition, to further assess robustness, we conduct an ablation study using different amounts of training data in Door and Pen environments: 90% (as in the paper), 75%, 50%, and 25%. The results below show that while the success rate decreases as the amount of training data is reduced (as expected), the trajectory similarity scores remain consistent. This demonstrates the robustness of our method.
>
> |||90% (22:3)|75% (18:7)|50% (12:13)|25% (6:19)|
> |-|-|-|-|-|-|
> |Door|DTW$_s$ (↓)|302.19|261.29|277.42|350.23|
> ||DTW$_a$ (↓)|275.20|257.20|263.66|335.20|
> ||WD$_s$ (↓)|4.59|4.38|4.42|5.19|
> ||WD$_a$ (↓)|3.73|3.37|3.27|4.10|
> ||Success|0.93|0.98|0.85|0.48|
> |||||||
> |Pen|DTW$_s$ (↓)|740.45|679.96|669.14|686.26|
> ||DTW$_a$ (↓)|664.75|629.19|624.25|654.41|
> ||WD$_s$ (↓)|8.55|8.02|7.79|8.24|
> ||WD$_a$ (↓)|6.05|5.95|5.53|5.85|
> ||Success|0.42|0.33|0.33|0.23|
>
> \* Note: DTW and WD are not normalized; lower is better.
>
> > W3: There doesn’t seem to be a clear observation that MAQ degrades task performance for the highest scoring RL algorithm, RLPD. Difference is even more obious when looking at the appendix reward curves, Figure 6.
> > Q2: RLPD Degradation: In Table 1 and Appendix Figure 6, RLPD seems to suffer significantly when combined with MAQ. Can the authors clarify this discrepancy between reported success rate and observed reward curves? Can you provide more details on how the success rate is computed vs reward curve normalized score
>
> We were also curious about why MAQ+RLPD performed worse than RLPD, especially in the Hammer environment, where the success rate drops from 1.00 to 0.56. After submitting the paper, we conducted a deeper analysis of this task. We find that in the 25 human demonstration trajectories, humans take an average of 451.4 steps to successfully complete the task. However, in the D4RL environment, the agent is restricted to a maximum of 200 steps per trajectory, after which the episode is automatically considered a failure.
>
> Since MAQ learns from human demonstrations, it tends to favor smooth but slow movements, while non-human-like RL agents tend to complete the task more quickly but less naturally. To validate this hypothesis, we increased the maximum episode length from 200 to 450 steps. As shown in the table below, MAQ+RLPD's success rate improves significantly from 0.56 to 0.72 while maintaining human-likeness scores. We will include this analysis with an explanation in the revision.
>
> ||RLPD|MAQ+RLPD (200)|MAQ+RLPD (450)|
> |-|-|-|-|
> |DTW$_s$ (↓)|845.88|578.82|648.32|
> |DTW$_a$ (↓)|1178.74|528.52|637.25|
> |WD$_s$ (↓)|9.57|5.20|6.17|
> |WD$_a$ (↓)|7.25|3.32|4.15|
> |Success|1.00|0.56|0.72|
>
> \* Note: DTW and WD are not normalized; lower is better.
>
> Regarding the success rate and normalized reward, in D4RL, the success rate indicates whether the agent successfully completes the task within the episode limit (e.g., 200 steps). In contrast, the normalized reward is a heuristic score defined by the D4RL environment to reflect how close the agent gets to completing the task. For example, in the Door task, the reward increases as the door gets closer to being fully opened. However, only when the door is completely opened, the task marked as successful. Overall, tasks like those in D4RL are primarily evaluated based on whether the agent actually succeeds since the rewards are often used to guide learning, but do not directly indicate task success. This is why we report and compare success rates in the main experiment.
>
> > W4: The motivation for human-like behavior is weakly argued and under-cited, beyond bringing in discussion the Turing Test.
>
> We understand the reviewer's concern. Indeed, the topic of human-like behavior in RL has been relatively underexplored, as most of the prior work has focused primarily on improving task performance rather than human-like behavior. This is why we start with the concept of the Turing Test to emphasize the importance of developing agents that are not only effective but also human-like. We will revise the paper to strengthen the motivation for human-like RL and include a Broader Impact section (see the response to W6). If the reviewer has any recommended citations or related works, we would be grateful for the suggestions.
>
> > W5: The number of human evaluators is not reported, and no statistical significance analysis is provided for the Turing Test or ranking results.
> > Q1: Statistical Validity of Human Evaluation: How many human evaluators participated in the Turing Test and ranking tests? Please include this number and report significance tests to support the reliability of these results.
>
> Thank you for pointing this out. In our human study, a total of 19 human evaluators participated in the Turing Test and ranking test. We will clarify this in the revision.
>
> In addition, we follow the similar method used in [1] reporting 96% confidence intervals for both the Turing Test and the ranking test. Thank you again for raising this point.
>
> [1] Tirinzoni, Andrea, et al. "Zero-shot whole-body humanoid control via behavioral foundation models." arXiv preprint arXiv:2504.11054 (2025).
>
> > W6: The paper lacks a Broader Impact or Societal Impact discussion, especially relevant when aiming to mimic human behavior.
> > Q4: Motivation and Impact: The paper briefly mentions the importance of human-likeness but does not provide a compelling argument or citations. Could the authors expand on the value and implications of designing human-like agents, including societal risks or limitations?
> > The paper does not adequately discuss the limitations or potential societal impacts of building human-like agents. The Discussion section does not meaningfully reflect on possible negative consequences or ethical concerns (e.g., manipulation, human trust, or deceptive agents). I recommend the authors include a Broader Impacts section that honestly addresses these issues and situates their work in a wider context.
>
> Thank you for the suggestion. We will include a Broader Impact section in the revision to address both the potential benefits and risks of human-like agents. The following are several key points we plan to cover. On the positive side, human-like behavior can offer practical benefits in real-world applications, particularly in scenarios involving human-robot collaboration. For example, a robot that assists with human-like movements can improve safety and trust, reducing the risk of accidents. On the other hand, human-like agents may be misused in deceptive or manipulative ways, such as cheating in competitive games. We will include these aspects in the revision. If reviewers feel there are additional aspects worth addressing, we would be happy to include them.
>
> > W7: It remains unclear whether demonstrations used in the human-likeness evaluation are successful ones. This could unfairly bias comparisons.
> > Q3: Demonstration Quality: Were the human demonstrations used in the evaluation (e.g., Turing Test) all successful executions? Were agent demonstrations sampled from successful rollouts? This information is crucial for fair comparison.
>
> Yes, all 25 human demonstrations used in our experiments are successful trajectories, as provided in [2]. We will clarify this in the revision. Thank you for pointing it out.
>
> [2] Rajeswaran, Aravind, et al. "Learning complex dexterous manipulation with deep reinforcement learning and demonstrations." arXiv preprint arXiv:1709.10087 (2017).

---

> > ### Comment · Reviewer_HJQk · 2025-08-05
> >
> > Thank you for the detailed rebuttal. I appreciate the clarifications, particularly regarding the evaluation protocol and the RLPD analysis. I look forward to seeing these improvements incorporated into the final version.

---

> ### Author Response · Authors · 2025-08-07
>
> Thank you for your replies and for the time and effort to review our paper and provide thoughtful suggestions. We are glad that our clarifications have addressed your concerns. We will make our best effort to incorporate these improvements into the final version.

---

### Official Review · Reviewer_rcnz · 2025-07-01

**Clarity:** 3
**Significance:** 2
**Originality:** 2
**Rating:** 4
**Confidence:** 4

**Summary:**

This paper proposes learning human-like RL agents to be explicable and facilitate human-AI interaction. The key idea is to generate behavior via connecting human-like behavior segments learned from human demonstrations.These segments are then viewed as options in RL. Generalization to different situations is achieved by the application of vector quantized variational autoencoder (VQVAE). The system is evaluated in four Adroit tasks (including opening door, hammering a nail, twirling a pen, moving a ball), where the generated behaviors are compared with human demonstrations using dynamic time warping and Wasserstein distances, success rate, as well as via human studies. Results show that the generated behaviors achieve higher similarity scores and success rates, while being less distinguishable from human demonstrations.

**Questions:**

1. Any reason for why prior works that leverage human demonstrations and can be used to generate "human-like" behaviors are not considered or compared, such as GAIL (see below)? Note that directly comparing with imitation learning that does not have the ability to adapt/generalize (such as BC) is not exactly fair ("This is because the BC agent learns to imitate state-action pairs from the demonstrations, without optimizing for successful outcomes.").

Ho, Jonathan, and Stefano Ermon. "Generative adversarial imitation learning." Advances in neural information processing systems 29 (2016).

2. Three different RL methods are chosen as the baselines to evaluate the generalizability of the proposed work. Why is this efficient use of space given that the learned macro actions are essentially treated as actions (assuming my understanding is correct) so it should generalize to whatever standard RL method? After all, we are not really interested in comparing these RL methods.

3. In general, there is significant variance in the performance across different tasks. What do you think the problem is and how you plan to address it?

4. In 5.3.2, it is claimed that "Moreover, MAQ+RLPD achieves a 71% win rate, nearly indistinguishable from the human demonstrations, which have a 74% win rate. This suggests that MAQ+RLPD can convincingly fool human evaluators into believing its behavior is human-like." It this true? Even though MAQ+RLPD has an overall win rate of 71%, Figure 4 still shows that human demonstration dominates it, no?

5. How do you set the reward for the macro actions? I did not find information about that in the paper.

**Ethical Concerns:**

["NO or VERY MINOR ethics concerns only"]

**Final Justification:**

Although I still maintain concerns regarding the fixed length macro actions and the use of RL to chain these seemingly "arbitrary" macro actions, the authors have provided reasonable evidence that demonstrated its effectiveness at generating human-like behaviors. To some extent, I was quite surprised at the results given the simplicity of the approach but kudos to the authors. Hence, I am adjusting my rating to "weak accept".

On the other hand, in addition to incorporating the relevant discussion during the author-reviewer discussion phase into the final version, I would encourage the authors to more carefully discuss the limitations of their approach, such as fixed length macro actions and potentially the difficulty with out of distribution scenarios. I would also encourage the authors to discuss the limitations of the evaluation. I feel that the performance improvement against the baselines could as well be explained away by other factors, such as whether or not the task objective is optimized (BC) or the computational advantage of using macro actions with RL (IQ-learn), not necessarily the proposed human-like behavior generation mechanism. Along this line, how would you expect MAQ to compare against RL with options where the options are manually constructed or learned? I would be curious to see the results.

**Limitations:**

Some of the limitations are discussed by the authors while others remain unaddressed. Please refer to my comments above.

**Paper Formatting Concerns:**

NA.

**Quality:**

2

**Strengths And Weaknesses:**

Strengths:

1. The paper is well motivated and clearly written.
2. The behavior generation pipeline seems to be effective in creating more human-like behaviors than the selected baselines.

Weaknesses:

1. The idea of using macro actions is restrictive for RL agents in contrast to prior work that may be leveraged to achieve similar results. Furthermore, the human-likeliness relies on the assumption of compositionality of macro actions, which may or may not hold in general.
2. Learning VQVAE requires expert demonstrations that are carefully mapped to critical subtasks to be generalizable.
3. Even though the proposed approach seems to have improved human likeliness with respect to the selected baselines, the result is highly domain dependent, revealing a critical gap to be filled.

---

> ### Author Rebuttal · Authors · 2025-07-31
>
> We thank the reviewer for constructive feedback and thoughtful reviews. We address each concern and question as follows.
>
> > W1: The idea of using macro actions is restrictive for RL agents in contrast to prior work that may be leveraged to achieve similar results. Furthermore, the human-likeliness relies on the assumption of compositionality of macro actions, which may or may not hold in general.
>
> We chose to use macro actions based on the assumption that sequences of actions better reflect human behavior. While prior work has attempted to produce human-like behavior using hand-crafted cost functions or reward shaping at the primitive action level, such approaches often require task-specific domain knowledge, limiting their generality. In contrast, our method relies only on human demonstrations and avoids manual task-specific design, making it more applicable across domains.
>
> To address concerns regarding potential restrictions in executing macro actions, we also conducted an ablation study in which agents still learn macro actions, but at test time only execute the first few steps of each macro action to allow more reactive behavior. Specifically, with a macro-action sequence length of 9, we tested executing only the first $j \in${$1,3,5,7$} actions. The table below shows that both human-likeness and performance remain similar across different $j$. Notably, even when $j=1$, the agent remains human-like while increasing reactivity.
>
> |j=9 (paper setting)|j=7|j=5|j=3|j=1|
> |-|-|-|-|-|
> |DTW$_s$ (↓)|740.45|722.54|730.76|738.15|730.31|730.31|
> |DTW$_a$ (↓)|664.75|651.99|664.43|666.73|673.95|673.95|
> |WD$_s$ (↓)|8.55|8.49|8.54|8.52|8.53|8.53|
> |WD$_a$ (↓)|6.05|6.00|6.05|6.03|6.09|6.09|
> |Success|0.42|0.37|0.44|0.41|0.42|0.42|
>
> \* Note: DTW and WD are not normalized; lower is better.
>
> Regarding the compositionality assumption, we agree that this is an assumption. However, our empirical results suggest that the learned macro actions are reasonably composable in practice. We also agree that a study of macro-action compositionality would be a valuable direction for future work.
>
> > W2: VQVAE requires expert demonstrations …
>
> We would like to clarify that our proposed MAQ framework does not require expert or optimal human demonstrations. Instead, MAQ is designed to distill human-like behaviors from any human demonstration and integrate them with RL algorithms to improve task performance.
>
> For example, in D4RL [1][2], two types of datasets are provided: (a) a human demonstration dataset (25 trajectories per task) collected from humans, and (b) an expert demonstration dataset (5000 trajectories per task) generated by a fine-tuned expert RL policy. As shown in the table below, the rewards of the human dataset are substantially lower than those of the expert dataset, indicating that the human demonstrations are far from optimal.
>
> |Task|Human dataset reward|Expert dataset reward|
> |-|-|-|
> |Door|784.9|2899.8|
> |Hammer|3049.7|12275.5|
> |Pen|6290.2|3283.5|
> |Relocate|3643.2|4293.5|
>
> In our experiments, MAQ is trained solely on the human dataset, without using any expert dataset. Although these trajectories are not optimal, our results show that MAQ is still able to achieve both human-likeness and high success rates. This shows that MAQ can effectively capture human behavior even from suboptimal demonstrations.
>
> [1] Rajeswaran, Aravind, et al. "Learning complex dexterous manipulation with deep reinforcement learning and demonstrations." arXiv preprint arXiv:1709.10087 (2017).
>
> [2] Fu, Justin, et al. "D4rl: Datasets for deep data-driven reinforcement learning." arXiv preprint arXiv:2004.07219 (2020).
>
> > W3: … the result is highly domain dependent …
>
> We agree with the reviewer that MAQ requires access to human demonstrations in the target domain, as noted in our Discussion section: "*Although MAQ exhibits more human-like behavior, one limitation is its reliance on human demonstrations to distill macro actions. The quality of these demonstrations can affect the effectiveness of MAQ. However, since our primary goal is to pursue human-likeness, obtaining human demonstrations is both necessary and appropriate within the scope of this work.*"
>
> However, we would like to emphasize that, compared to prior human-like RL approaches that rely on domain knowledge to manually design human-likeness cost functions, our method only requires collecting demonstrations. This greatly reduces the effort and cost of adapting to new domains, making MAQ more practical for real-world applications. While extending MAQ to cross-domain generalization remains an interesting direction for future work, we believe our method already represents a significant step in human-like RL by offering a general, data-driven solution that avoids handcrafted, domain-specific constraints.
>
> > Q1: Any reason for why prior works that leverage human demonstrations and can be used to generate "human-like" behaviors are not considered or compared, such as GAIL (see below)?
>
> We include BC as a baseline to represent a simple and widely used method. As the reviewer noted, BC captures human-likeness to some degree but does not optimize with task success rate. We did not include GAIL for several reasons. First, GAIL relies on high-quality expert demonstrations, while our focus is on imitating human behavior, which can be suboptimal but diverse. Second, GAIL is known to require extensive training time to train in high-dimensional continuous control tasks. For example, training GAIL on the 6-dimensional Walker task takes ~25M steps. In contrast, our method learns macro actions for the 24-dimensional Pen task in only 1M steps. We also attempted to train GAIL for 1M steps on the four D4RL tasks, but GAIL failed to produce any successful trajectories, possibly requiring more training and computational resources.
>
> However, we agree that comparing other imitation learning methods would strengthen our results. To this end, we included IQ-Learn [1], one of the strongest IL algorithms. Since IQ-Learn was not originally tested on D4RL tasks, we applied it (using the hyperparameters recommended in the original paper) to all four tasks. The results below show that IQ-Learn performs poorly in both human-likeness and task success, while MAQ+IQ-Learn improves both performance. We believe this additional experiment offers a more convincing comparison and also demonstrates that MAQ is complementary to IL algorithms. We can include this experiment if the reviewer finds it satisfactory.
>
> [1] Garg, Divyansh, et al. "Iq-learn: Inverse soft-q learning for imitation." Advances in Neural Information
>
> |||IQ-Learn|MAQ+IQ-Learn|
> |-|-|-|-|
> |Door|DTW$_s$ (↓)|633.88|358.11|
> ||DTW$_a$ (↓)|796.35|345.33|
> ||WD$_s$ (↓)|9.10|5.11|
> ||WD$_a$ (↓)|8.18|4.04|
> ||Success|0.00|0.25|
> ||
> |Hammer|DTW$_s$ (↓)|844.00|627.05|
> ||DTW$_a$ (↓)|1261.88|607.84|
> ||WD$_s$ (↓)|9.58|6.25|
> ||WD$_a$ (↓)|7.41|4.18|
> ||Success|0.00|0.04|
> ||
> |Pen|DTW$_s$ (↓)|1025.75|778.53|
> ||DTW$_a$ (↓)|768.69|684.22|
> ||WD$_s$ (↓)|10.32|8.86|
> ||WD$_a$ (↓)|7.11|6.38|
> ||Success|0.09|0.20|
> ||
> |relocate|DTW$_s$ (↓)|641.08|534.19|
> ||DTW$_a$ (↓)|1261.29|591.09|
> ||WD$_s$ (↓)|10.20|7.72|
> ||WD$_a$ (↓)|8.77|5.62|
> ||Success|0.00|0.02|
>
> \* Note: DTW and WD are not normalized; lower is better.
>
> > Q2: Three different RL methods are chosen … Why is this efficient use of space given that the learned macro actions are essentially treated as actions (assuming my understanding is correct) so it should generalize to whatever standard RL method?
>
> We fully agree with the reviewer's point! As noted, MAQ is fundamentally a framework that transforms actions into learned macro actions, making it to be easily integrated into various RL algorithms. However, we were concerned that reviewers/readers might expect empirical evidence of this generality. Therefore, we included experiments with three representative RL algorithms, IQL, SAC, and RLPD, to demonstrate that MAQ consistently improves human-likeness while maintaining high task success rates across different RL algorithms. Note that our intention is not to compare these RL algorithms themselves, but to show that each benefits from integrating MAQ, thereby highlighting its flexibility and generality. We welcome any suggestions for improving our experimental design.
>
> > Q3: … significant variance in the performance across different tasks.
>
> The performance variance is primarily due to inherent differences in task difficulty. For example, the D4RL paper [2] (Table 2) reports that among the four Adroit tasks, Pen and Door are relatively easier, while Relocate (requiring precise grasping, lifting, and moving of an object) is the most challenging. This result also aligns with our experimental results. In summary, the performance variance is not specific to MAQ, but is based on differences in task difficulty.
>
> > Q4: Even though MAQ+RLPD has an overall win rate of 71%, Figure 4 still shows that human demonstration dominates it, no?
>
> It is true that when directly comparing human demonstrations against MAQ+RLPD in the ranking test, human demonstrations have a 62% win rate. However, our original claim is based on the **overall win rate across all agent pairs**. For example, human demonstrations only achieve an 87% win rate against SAC, whereas MAQ+RLPD achieves a 95% win rate against SAC. This suggests that human evaluators found MAQ+RLPD to be more human-like than human demonstrations when both are compared to SAC. Therefore, our intention is to highlight MAQ+RLPD's overall performance in fooling human evaluators across all comparisons, rather than to claim it outperforms human demonstrations directly. We thank the reviewer for the feedback and will revise the phrasing in the revision to clarify this more precisely.
>
> > Q5: How do you set the reward for the macro actions?
>
> The reward of macro action is computed as the sum of each step's reward over the entire action sequence. We will include this information in the revision.

---

> > ### Comment · Reviewer_rcnz · 2025-08-04
> >
> > Thank you for the detailed response. Here is my general feeling after the rebuttal:
> >
> > Concerns remained:
> >
> > 1) The compositionality of the macro actions remains a strong assumption.
> > 2) The availability of human demonstrations as macro actions remains a strong assumption.
> >
> > Concerns alleviated:
> >
> > 1) Baselines: The additional results on IQL is helpful. However, can you explain the low success rates, as well as why the magnitude of results seems to differ much from that reported in the paper (Table I)?

---

> > > ### Author Response · Authors · 2025-08-05
> > >
> > > Thank you for your feedback.
> > >
> > > > The availability of human demonstrations as macro actions remains a strong assumption
> > >
> > > First, since the primary goal is to produce human-like behavior, the use of human demonstrations is not only reasonable but necessary to define what constitutes human-likeness. Without access to human demonstrations, it would be difficult to quantify human-like behavior in a principled way.
> > >
> > > Second, collecting human or expert demonstrations is actually quite common in Learning from Demonstration (LfD) and imitation learning (IL) for robot control. Specifically, many prior works on LfD or IL rely on human operators to provide demonstrations for learning, such as RT-1 [1], RT-2 [2], Robomimic [3], and OpenVLA [4]. If LfD and IL from demonstrations of human experts are regarded as feasible and practical frameworks, then the availability of human demonstrations (not necessarily of expert level) for learning macro actions shall also be a mild requirement (at least no stronger than the demonstration requirements of LfD and IL).
> > >
> > > Moreover, we highlight that our method only needs a fairly small number of human demonstrations. As shown in our rebuttal (cf. response to W2 for Reviewer HJQk), we conducted additional experiments using different amounts of training data in Door and Pen environments: 90% (as in the paper), 75%, 50%, and 25%.
> > >
> > > The results below show our method can still achieve human-like behavior even when trained with **as few as 6 human demonstrations**. This further suggests that the requirement of human demonstrations under our method is indeed fairly mild.
> > >
> > > |||90% (22:3)|75% (18:7)|50% (12:13)|25% (6:19)|
> > > |-|-|-|-|-|-|
> > > |Door|DTW$_s$ (↓)|302.19|261.29|277.42|350.23|
> > > ||DTW$_a$ (↓)|275.20|257.20|263.66|335.20|
> > > ||WD$_s$ (↓)|4.59|4.38|4.42|5.19|
> > > ||WD$_a$ (↓)|3.73|3.37|3.27|4.10|
> > > ||Success|0.93|0.98|0.85|0.48|
> > > |||||||
> > > |Pen|DTW$_s$ (↓)|740.45|679.96|669.14|686.26|
> > > ||DTW$_a$ (↓)|664.75|629.19|624.25|654.41|
> > > ||WD$_s$ (↓)|8.55|8.02|7.79|8.24|
> > > ||WD$_a$ (↓)|6.05|5.95|5.53|5.85|
> > > ||Success|0.42|0.33|0.33|0.23|
> > >
> > > \* Note: DTW and WD are not normalized; lower is better.
> > >
> > > [1] Brohan, Anthony, et al. "RT-1: Robotics transformer for real-world control at scale." RSS 2023.
> > >
> > > [2] Zitkovich, Brianna, et al. "RT-2: Vision-language-action models transfer web knowledge to robotic control." CoRL 2023.
> > >
> > > [3] Mandlekar, Ajay, et al. "What Matters in Learning from Offline Human Demonstrations for Robot Manipulation." CoRL 2022.
> > >
> > > [4] Kim, Moo Jin, et al. "OpenVLA: An Open-Source Vision-Language-Action Model." CoRL 2025.
> > >
> > > > The compositionality of the macro actions remains a strong assumption.
> > >
> > > The goal of the VQVAE is to learn representative segments (macro actions) from human demonstrations, which inherently contain the behaviors necessary to complete the task. Namely, each macro action corresponds to a behavior fragment that is already known to contribute to task success.
> > >
> > > Therefore, composing these learned macro actions for success is not based on a strong assumption, but rather on the fact that they are extracted from demonstrations that achieve the task. If the VQVAE successfully captures behavior segments, the RL agent can learn to recombine them effectively to produce new trajectories that complete the task.
> > >
> > > > Can you explain the low success rates, as well as why the magnitude of results seems to differ much from that reported in the paper (Table I)?
> > >
> > > The low success rates of IQ-Learn in the Adroit environment can be attributed to its reliance on expert demonstrations, as well as the high-dimensional action space and sparse reward, which increase the difficulty of training. These results are consistent with those reported in [5].
> > >
> > > Note that we observe that applying the MAQ framework improves success rates. In addition, the similarity scores (DTW/WD) provided in our previous response were unnormalized. Below, we provide the normalized scores, which are directly comparable to those in Table I of our paper.
> > >
> > > |||IQ-Learn|MAQ+IQ-Learn|
> > > |-|-|-|-|
> > > |Door|DTW$_s$|0.02|0.63|
> > > ||DTW$_a$|0.31|0.83|
> > > ||WD$_s$|-0.12|0.61|
> > > ||WD$_a$|0.09|0.72|
> > > ||Success|0.00|0.25|
> > > |||||
> > > |Hammer|DTW$_s$|-0.02|0.55|
> > > ||DTW$_a$|0.29|0.87|
> > > ||WD$_s$|-0.03|0.57|
> > > ||WD$_a$|0.17|0.71|
> > > ||Success|0.00|0.04|
> > > |||||
> > > |Pen|DTW$_s$|-0.17|0.45|
> > > ||DTW$_a$|0.25|0.52|
> > > ||WD$_s$|0.31|0.54|
> > > ||WD$_a$|0.44|0.59|
> > > ||Success|0.09|0.20|
> > > |||||
> > > |relocate|DTW$_s$|0.12|0.34|
> > > ||DTW$_a$|0.28|0.76|
> > > ||WD$_s$|0.08|0.42|
> > > ||WD$_a$|0.22|0.62|
> > > ||Success|0.00|0.02|
> > >
> > > [5] Zhang, Wenjia, et al. "Discriminator-guided model-based offline imitation learning." CoRL, 2023.
> > >
> > > We will also incorporate the above discussions in the revision. We hope the reviewer will kindly reconsider the evaluation based on the clarifications we provided above. Again, we thank the reviewer for all the constructive feedback and the time the reviewer put into helping us improve our submission.

---

> > > > ### Comment · Reviewer_rcnz · 2025-08-05
> > > >
> > > > Thank you for the clarification regarding the IQ-Learn results. I will explain my remaining concerns in more detail below:
> > > >
> > > > 1. My first concern was about the assumption of the availability of human demonstrations AS MACRO ACTIONS, not the availability of human demonstrations in general. From Figure 1, it appears that MAQ requires the macro action and its associated action sequence to be available. This is a strong assumption since it requires the identification of macro actions (that are reusable and composable) and then soliciting demonstrations for these macro actions. For example, option learning has been proven to be challenging since it must first deal with the segmentation of action sequences into meaningful options.
> > > >
> > > > 2. My concern about the compositionality was due in part to the use of the RL agent. Since RL agent focuses on reward optimization, it tends to ignore the "context" that is inherent in the execution of the macro actions, especially when they may differ from demonstrations due to environment and task perturbations. For instance, a slight change in the environment (e.g., a delicate obstacle) may result in significant changes to the behavior to maintain human-likeness.

---

> ### Author Response · Authors · 2025-08-07
>
> We thank the reviewer for the further clarification and for engaging in the discussion!
> > My first concern was about the assumption of the availability of human demonstrations AS MACRO ACTIONS, not the availability of human demonstrations in general. …
>
> > This is a strong assumption since it requires the identification of macro actions (that are reusable and composable) and **then** soliciting demonstrations for these macro actions. …
>
> We thank the reviewer for the clarification, and we would like to clarify a possible misreading of our method (we apologize if we misunderstood the reviewer's concern). Our approach does not require identifying macro actions first and then collecting human demonstrations specifically for those predefined macro actions. Instead, we extract fixed-length segments (9 in the paper) from the human demonstrations. Given each state $s$ in a demonstration trajectory, we define a macro action $m$ as the sequence of 9 consecutive primitive actions starting at $s$ (see Figure 1).  As a result, every state in the training data is associated with a macro action (a human-like segment). In summary, we **first** collect human demonstrations, and **then** macro actions are derived from these trajectories.
>
> Regarding whether these macro actions are reusable and composable, as mentioned in our previous response: since they are extracted from human demonstrations, they inherently contain the behaviors necessary to complete the task.
>
> We also agree with the reviewer that using segmentation methods from option learning [1, 2, 3] to determine variable-length macro actions per state is an interesting direction. However, in the work, our primary focus is not on the segmentation strategy itself, but rather on **demonstrating that macro actions derived directly from human demonstrations** can allow RL to achieve both a high task success rate and human-likeness. We appreciate this suggestion and will include it in our discussion of future work.
>
> [1] Kipf, Thomas, et al. "CompILE: Compositional imitation learning and execution." ICML 2019.
>
> [2] Chang, Yi-Hsiang, et al. "Reusability and transferability of macro actions for reinforcement learning." ACM TELO 2022.
>
> [3] Konidaris, George, et al. "Constructing skill trees for reinforcement learning agents from demonstration trajectories." NeurIPS 2010.
>
> > My concern about the compositionality was due in part to the use of the RL agent. …
>
> Our current method is developed under the standard RL setting where the human demonstrations and the learned RL agents operate within the same MDP. Namely, the environment dynamics and the task specification are identical.
>
> We agree with the reviewer that handling environment perturbations or generalizing across different environments is a very interesting direction. To achieve such generalization, the MAQ framework could be extended to a meta-RL setting [4, 5], where the agent is provided with a distribution over several tasks during training (e.g., tasks involving obstacle avoidance), along with demonstrations from each task. This setup would allow the agent to learn reusable and adaptive macro actions across multiple tasks. In this way, the agent can generalize to similar but unseen tasks during testing (e.g., an environment with an unseen delicate obstacle). We appreciate this suggestion and will include it in our discussion of future work.
>
> [4] Finn, Chelsea, Pieter Abbeel, and Sergey Levine. "Model-agnostic meta-learning for fast adaptation of deep networks." ICML 2017.
>
> [5] Rakelly, Kate, et al. "Efficient off-policy meta-reinforcement learning via probabilistic context variables." ICML 2019.
>
> We hope the above explanation addresses the reviewer's concerns, and we would be happy to engage in any further discussion to improve our submission.

---

> > ### Comment · Reviewer_rcnz · 2025-08-07
> >
> > Thank you for the additional explanation. I would like the authors to clarify the following:
> >
> > You mentioned "we extract fixed-length segments (9 in the paper) from the human demonstrations". Do you mean that you use EVERY length-9 segment in the human demonstrations for training, or that you choose SOME length-9 segments? If the former, I would be concerned about fixing the length of the macro actions (which can result in jumbled macro actions). If the latter, how do you decide which ones? Learning macro actions this way requires domain knowledge and will severely limit the applicability of MAQ.
> >
> > Limiting the length of the macro actions can also damage the reusabiity/compositionality of the macro actions, which could result in MAQ only working with simple tasks (e.g., tasks that require only a single macro action). It would face major challenges dealing with complex tasks that require the chaining of multiple macro actions (of potentially different lengths). Have you investigated in this point?

---

> > > ### Author Response · Authors · 2025-08-08
> > >
> > > We thank the reviewer for the feedback.
> > >
> > > * Marco-action extraction
> > >
> > > We use the former. EVERY length-9 segment in the human demonstrations is used to train the VQVAE. Below, We address the reviewer's concerns regarding jumbled macro actions, reusability, and compositionality in our MAQ framework.
> > >
> > > * jumbled macro actions
> > >
> > > MAQ uses a **state-conditioned Conditional-VQVAE**. For each state, the VQVAE encodes and decodes the corresponding sequence of actions, where these macro actions are specific to that state. Since all training data come from human demonstrations, each learned macro action reflects a human-like behavior for the given state. Note that different states produce different macro actions; these macro actions are not shared across unrelated states, but are related to their originating state, so they are not "jumbled".
> > >
> > > * reusability
> > >
> > > In option learning, reusability is often emphasized to discover options that can be applied across diverse situations (states). In contrast, MAQ focuses on state-conditioned macro actions: if a macro action happens to be applicable to multiple states, the VQVAE will output the same macro action when given those states, without explicitly transferring it from one state to another. Therefore, our framework does not require macro actions to be universally reusable across all states.
> > >
> > > * compositionality
> > >
> > > > It would face major challenges dealing with complex tasks that require the chaining of multiple macro actions.
> > >
> > > Because the VQVAE produces several macro actions for each state, **the RL agent's goal is to select and chain these state-conditioned macro actions to maximize both task success and human-likeness**. Our experiments demonstrate that MAQ achieves high task success rates, indicating RL can effectively handle the compositionality of these macro actions.
> > >
> > > > MAQ only working with simple tasks (e.g., tasks that require only a single macro action)
> > >
> > > We respectfully disagree that MAQ is limited to simple tasks. The Adroit suite is a sparse-reward benchmark that requires controlling at least 24 degrees of freedom in the ShadowHand environment. For example, the pen environment consists of a 28-dimensional continuous action space. These make Adroit a relatively challenging task, rather than a simple task.
> > >
> > > Finally, we appreciate the suggestion to explore macro actions of varying lengths. For example, one possible direct extension is to apply MAQ with multiple VQVAEs, each of which extracts macro actions of a specific length, and then consider the union of all macro actions of different lengths for RL. We agree this is a promising extension and will include it in our discussion of future work.
> > > We hope the above explanation addresses the reviewer's concerns and will kindly reconsider the evaluation based on the clarifications we provided above.

---

> > > > ### Comment · Reviewer_rcnz · 2025-08-08
> > > >
> > > > Thank you for the clarifications. Can you please also report here the average number of actions (i.e., length of the action sequence, NOT the number of macro actions) for each of the Adroit tasks?

---

> > > > > ### Author Response · Authors · 2025-08-08
> > > > >
> > > > > Sure, below we report the average length (number of actions) in human demonstration trajectories for each task. (Each task consists of 25 trajectories.)
> > > > >
> > > > > |Tasks|Average length in trajectories|
> > > > > |-|-|
> > > > > |Door|268.16|
> > > > > |Hammer|451.4|
> > > > > |Pen|199.0|
> > > > > |Relocate|396.68|
> > > > >
> > > > > Please let us know if any additional information would be helpful.

---

### Official Review · Reviewer_HZYt · 2025-07-03

**Clarity:** 4
**Significance:** 3
**Originality:** 4
**Rating:** 5
**Confidence:** 4

**Summary:**

- The paper explores the question of how to design agents that accomplish tasks like humans do.
- The paper tries to generate more human-like and relatively smooth trajectories while avoiding traits like shakiness, and other unnatural behaviors.
- The authors introduce MAQ (Macro-Action Quantization) framework where human action sequences are distilled into macro-actions (sequence of actions) using a conditional VQ-VAE with an aim to match the generated action sequences with human-like demonstration along with maximizing the required rewards.
  - MAQ (state-conditioned VQ-VAE) transforms that primitive low-level action space into a high-level behavior-based macro actions space.
  - Enforces the policy to select these macro-actions that can be decoded back to more human line action sequences (trajectories).
  - The VQ-VAE codebook, on a high-level, represents a collection of human-like meta-actions or behavior or skills that can be combined together to form a trajectory. It becomes a reduced macro-action-space as compared to the original action space.
- The RL agent tries to optimize (in a semi-MDP setting) over short action sequences to capture detailed human behavior.
- The trajectories within an horizon (of sequence len = H) are executed based on receding-horizon control and are replanned after k (<= H) steps.
- The paper uses Dynamic Time Warping and Wasserstein distance as trajectory similarity metrics to calculate human-likeness in the generated trajectories.
- Proposes an interesting work to extend research on designing human-like agents/trajectories.

**Questions:**

- How are we making sure that we have proper coverage of the state-action pair for human demonstrations for different experiments to have effective/generalized quantization?
- Any particular way/frequency of updating the codebook (clusters) or having more robust codebooks that don't change drastically with the new information.
- Line 123: Are there any ablations for j to have reactivity vs human-likeness metrics comparison?
- Equation 4: What is this loss term $||m - \tilde{m}||^2$ ? Is this the L2 norm over the action sequences or log-likelihood?


**Suggestion:**
It seems that this framework can also be leveraged for hierarchical RL with the high-level (low frequency) behaviour planner and a low-level (high-frequency planner).

**Ethical Concerns:**

["NO or VERY MINOR ethics concerns only"]

**Final Justification:**

Addressed Concerns:
- The authors have addressed the concern related to dynamic environments.
- They conducted experiments to test the proper coverage of the state-action pairs.
- They conducted experiments/ablations for j to have reactivity vs human-likeness metrics comparison.
- Compared the results with other IL algorithms like IQL.
- The authors successfully clarified that the MAQ framework does not require optimal or expert demonstrations.
- The use of codebook reinitialization was explained as a method to ensure the robustness and effective usage of the codebook.

It seems that the quality of the paper can be/would be improved after the rebuttal discussions and experiments are included in the paper. I would like to go my initial score of 4 considering the limitations pointed out in the overall rebuttal.

**Limitations:**

Yes.

**Paper Formatting Concerns:**

- Line 28: Typo in “making them they easily….”

**Quality:**

3

**Strengths And Weaknesses:**

**Strengths:**
- The paper does a good job of having an exhaustive analysis of the experiments based on the mathematical and human-evaluation metrics.
- The authors very well presented answers to the questions:
  - Can MAQ accomplish tasks while aligning with human demonstrations?
  - Can and how likely MAQ generated trajectories fool human evaluators?
- Presents different evaluation methods to showcase the results.
  - Turing test with human participants to distinguish between the generated (using different RL baselines) and human trajectories and had the following results:
   - MAQ+RLPD (39%) > MAQ+SAC (34%) > 288 MAQ+IQL (32%) > BC (24%) = RLPD (24%) > SAC (19%) > IQL (13%).
  - For the human-likeness ranking test: MAQ+RLPD achieved a 71% win rate, very close to the human demonstrations which had a 74% win rate.
  (*References for these baselines in the paper)

- The authors claimed the MAQ seems to have more potential to solve tasks requiring intricate scenarios with numerous macro actions.
- The results demonstrated that MAQ based agents can effectively complete the tasks and are human-like as well.


**Weaknesses/Limitations**:
- The proposed framework would require expert human or optimal demonstration data in order to encode near-human-like behaviors.
- The paper assumes that the human-demo are optimal trajectories and in that case, it might miss the intent of the task if they are near-suboptimal.
- The paper does not talk about any dynamic environments where the frequency of executing actions might play an important role in having more reactive policies and yet human-like.

Multiple aspects of this work can be derived for other tasks related to human-like task completion or human-like trajectory classification, or distilling/enforcement of particular conditional behaviors.

---

> ### Author Rebuttal · Authors · 2025-07-31
>
> We thank the reviewer for constructive feedback and thoughtful reviews. We address each concern and question as follows.
>
> > W1: The proposed framework would require expert human or optimal demonstration data in order to encode near-human-like behaviors
>
> We would like to clarify that our proposed MAQ framework does not require expert or optimal human demonstrations. Instead, MAQ is designed to distill human-like behaviors from any human demonstration and integrate them with RL algorithms to improve task performance.
>
> For example, in D4RL [1][2], two types of datasets are provided: (a) a human demonstration dataset (25 trajectories per task) collected from humans, and (b) an expert demonstration dataset (5000 trajectories per task) generated by a fine-tuned expert RL policy. As shown in the table below, the rewards of the human dataset are substantially lower than those of the expert dataset (except Pen), indicating that the human demonstrations are far from optimal.
>
> |Task|Human dataset reward|Expert dataset reward|
> |-|-|-|
> |Door|784.9|2899.8|
> |Hammer|3049.7|12275.5|
> |Pen|6290.2|3283.5|
> |Relocate|3643.2|4293.5|
>
> In our experiments, MAQ is trained solely on the human dataset, without using any expert dataset. Although these trajectories are not optimal, our results show that MAQ is still able to achieve both human-likeness and high success rates. This shows that MAQ can effectively capture human behavior even from suboptimal demonstrations.
>
> [1] Rajeswaran, Aravind, et al. "Learning complex dexterous manipulation with deep reinforcement learning and demonstrations." arXiv preprint arXiv:1709.10087 (2017).
>
> [2]   Fu, Justin, et al. "D4rl: Datasets for deep data-driven reinforcement learning." arXiv preprint arXiv:2004.07219 (2020).
>
> > W2: The paper assumes that the human-demo are optimal trajectories and in that case, it might miss the intent of the task if they are near-suboptimal.
>
> Following our previous response, MAQ does not require optimal trajectories. However, we agree with the reviewer that training on suboptimal demonstrations may affect task performance. We have also noted this in the Discussion section: "The quality of these demonstrations can affect the effectiveness of MAQ."
>
> To investigate the impact of demonstration quality, we conduct an ablation study using different subsets of trajectories with the lowest task rewards. Specifically, we split the training/testing dataset based on the reward of each trajectory and evaluate three scenarios:
> * 75% low-reward: train on 75% lowest-reward trajectories and test on the remaining 25% highest-reward ones
> * 50% low-reward: train on 50% lowest-reward trajectories and test on the remaining 50%
> * 25% low-reward: train on 25% lowest-reward trajectories and test on the remaining 75%
>
> Since reducing the size of the training dataset will affect the performance, we also include an experiment using the same data proportions but with trajectories selected randomly. The results are shown below.
>
> ||||75%|75%|50%|50%|25%|25%|
> |-|-|-|-|-|-|-|-|-|
> |||MAQ+RLPD|random|lowest|random|lowest|random|lowest|
> |Door|DTW$_s$ (↓)|302.19|261.29|239.77|277.42|298.26|350.23|414.27|
> ||DTW$_a$ (↓)|275.20|257.20|242.36|263.66|294.37|335.20|408.44|
> ||WD$_s$ (↓)|4.59|4.38|4.70|4.42|4.77|5.19|5.85|
> ||WD$_a$ (↓)|3.73|3.37|3.85|3.27|3.77|4.10|4.49|
> ||Success|0.93|0.98|0.79|0.85|0.55|0.48|0.03|
> ||
> |Pen|DTW$_s$ (↓)|740.45|679.96|590.02|669.14|649.56|686.26|686.74|
> ||DTW$_a$ (↓)|664.75|629.19|560.92|624.25|607.43|654.41|652.16|
> ||WD$_s$ (↓)|8.55|8.02|9.70|7.79|8.47|8.24|8.61|
> ||WD$_a$ (↓)|6.051|5.95|8.42|5.53|6.01|5.85|6.24|
> ||Success|0.417|0.33|0.32|0.33|0.23|0.23|0.20|
>
> \* Note: DTW and WD are not normalized; lower is better.
>
> As expected, training with fewer demonstrations reduces performance in both low-reward and random settings. Moreover, training on low-reward trajectories consistently leads to lower success rates compared to random. This demonstrates that the quality of demonstrations affects task performance. However, DTW and WD scores remain consistent across all scenarios, even when training on the lowest-reward trajectories. This suggests that MAQ can still effectively capture human behavior from suboptimal trajectories. We would be happy to include this additional ablation study in the revision if the reviewer finds it satisfactory.
>
> > W3: The paper does not talk about any dynamic environments where the frequency of executing actions might play an important role in having more reactive policies and yet human-like.
>
> To evaluate using MAQ in dynamic environments, we conduct an additional experiment on Enduro, a dynamic Atari environment with rapidly changing scenes, frequent decision-making, and the need for real-time reactions to avoid collisions.
>
> We train both PPO and MAQ+PPO on Enduro. For MAQ, the macro-action sequence length is set to 9. To assess human-likeness in the discrete action space, we adopt the method proposed in [3], which calculates the Wasserstein distance between action distributions, where xN represents the aggregated action histogram over N steps. The results below show that MAQ+PPO outperforms PPO in both scores and human-likeness, demonstrating that MAQ can effectively operate in dynamic environments while preserving human-like behavior.
>
> ||PPO|MAQ+PPO|
> |-|-|-|
> |WD$_a$ x1 (↓)|0.86|**0.45**|
> |WD$_a$ x8 (↓)|2.49|**1.55**|
> |WD$_a$ x16 (↓)|3.55|**2.59**|
> |Score|25.66|**425.33**|
>
> \* Note: DTW and WD are not normalized; lower is better.
>
> [3] Pearce, Tim, et al. "Imitating human behaviour with diffusion models." arXiv preprint arXiv:2301.10677 (2023).
>
> > Q1: How are we making sure that we have proper coverage of the state-action pair for human demonstrations for different experiments to have effective/generalized quantization?
>
> To investigate the coverage of state-action pairs from human demonstrations, we analyze whether actions from testing trajectories appear within the VQVAE codebook. Specifically, for each state-macro action pair $(s, m)$ from the testing trajectories, we input the state $s$ into the trained VQVAE decoder to reconstruct all macro actions corresponding to the codebook entries. Then, we check whether the macro action $m$ appears among these decoded outputs.
>
> Since D4RL uses continuous actions, we calculate the L2 distance between ground-truth macro action $m$ and each decoded macro action $\tilde{m}$. If the distance to any codebook entry is below a predefined threshold $t$, we consider $m$ to be covered by the codebook. The table below shows the evaluation results. For comparison, we also applied the same analysis to expert trajectories, which do not exhibit non-human-like behavior. The results demonstrate that our VQVAE trained on human demonstrations achieves high coverage on human trajectories but very low coverage on expert trajectories.
>
> |Threshold ($t$)|Human|Expert|
> |-|-|-|
> |0.1|98.77|52.33|
> |0.05|95.34|0.13|
> |0.04|93.84|0.008|
> |0.03|90.70|0|
> |0.02|82.22|0|
> |0.01|29.19|0|
>
> > Q2: Any particular way/frequency of updating the codebook (clusters) or having more robust codebooks that don't change drastically with the new information.
>
> We adopt a commonly used technique in VQVAE: codebook reinitialization [4], which helps maintain effective codebook usage and avoids codebook collapse. Specifically, if certain codebook entries remain unused for a period, they are reinitialized with the most frequently used embeddings. This strategy ensures that the codebook remains robust.
>
> [4] Williams, Will, et al. "Hierarchical quantized autoencoders." Advances in Neural Information Processing Systems 33 (2020): 4524-4535.
>
> > Q3: Line 123: Are there any ablations for j to have reactivity vs human-likeness metrics comparison?
>
> We conduct an experiment in the Pen environment using MAQ+RLPD with different values of $j$, including 1, 3, 5, and 7. The table below shows that both human-likeness and performance remain similar across different $j$. This is likely because $H$ (action sequence length) is set to 9 during MAQ training, so even when $j=1$, the agent is still able to execute human-like behavior.
>
> We choose $j=H$ in the paper because longer sequences tend to generate smoother, uninterrupted human-style motion. In addition, this setting reduces the replanning frequency, which improves training efficiency.
>
> |j=9 (paper setting)|j=7|j=5|j=3|j=1|
> |-|-|-|-|-|
> |DTW$_s$ (↓)|740.45|722.54|730.76|738.15|730.31|730.31|
> |DTW$_a$ (↓)|664.75|651.99|664.43|666.73|673.95|673.95|
> |WD$_s$ (↓)|8.55|8.49|8.54|8.52|8.53|8.53|
> |WD$_a$ (↓)|6.05|6.00|6.05|6.03|6.09|6.09|
> |Success|0.42|0.37|0.44|0.41|0.42|0.42|
>
> \* Note: DTW and WD are not normalized; lower is better.
>
> > Q4: Equation 4: What is this loss term ||m-~m||^2? Is this the L2 norm over the action sequences or log-likelihood?
>
> We use the L2 norm over the action sequences.

---

> > ### Comment · Reviewer_HZYt · 2025-08-07
> >
> > Thanking the authors for addressing the questions.
> > I went through the comments and discussions from other reviewers as well and it seems that the authors have tried to carefully address their concerns - some of which were shared across other reviewers.
> > - The authors have addressed the concern related to dynamic environments.
> > - They conducted experiments to test the proper coverage of the state-action pairs.
> > - They conducted experiments/ablations for j to have reactivity vs human-likeness metrics comparison.
> > - Compared the results with other IL algorithms like IQL.
> >
> > The quality of the paper can be improved if the  rebuttal discussions and experiments are included in the paper.
> > Appreciation for the authors was putting efforts in this direction and releasing their work. This work is a good combination of research and practical engineering. I would like to maintain my score considering some limitations pointed in the overall rebuttal.
> >
> > Best wishes! :)

---

> > > ### Author Response · Authors · 2025-08-07
> > >
> > > Thank you very much for the time and effort to review our paper and provide thoughtful suggestions to improve our submission. We will definitely make every effort to incorporate the discussions and experiments from the rebuttal into the final version.

---

### Comment · Area_Chair_LUak · 2025-08-04

Dear Reviewers,

Thank you for your thorough evaluation. Please take time to review the authors' responses to the concerns and questions you raised. If any points in their rebuttal require clarification or if you have additional questions, please provide your comments accordingly. Thank you.

Best,

AC

---

> ### Author Response · Authors · 2025-08-04
>
> Dear AC and Reviewers,
>
> Thank you for your efforts and time in handling and reviewing our papers. We are willing to clarify any further questions or concerns during the remaining author-reviewer period.
>
> Sincerely,
>
> Authors

---

### Note · Authors · 2025-08-15

We sincerely thank the reviewers for their valuable comments and for engaging in constructive discussions. Below, we summarize our contributions and common questions from reviewers:
## [Our Contributions]
- Our work is the first to achieve human-like RL through MPC combined with MAQ.
- We report both quantitative trajectory-similarity metrics and qualitative human studies (Turing Test and human-likeness ranking).
- Experiments show that MAQ can be easily integrated with various RL algorithms and empirically improves human-likeness across tasks.
## [Common Questions from Reviewers]
During the discussion phase, we conducted additional experiments and analyses addressing reviewer concerns:
- **Imitation Learning baseline (Reviewer rcnz, Reviewer gwPt)**: We included IQ-Learn as an additional imitation learning baseline. Results show that MAQ consistently outperformed IQ-Learn, highlighting its advantage over standard IL methods.
- **Generality to other environments (Reviewer HZYt, Reviewer gwPt)**: We applied MAQ to two Atari games (Enduro and Venture). Results show that MAQ+PPO outperformed PPO in both games, demonstrating that MAQ generalizes beyond continuous control tasks.
- **Reactivity (Reviewer HZYt, Reviewer rcnz)**: We added an ablation study on executing only the first $j$ actions to access reactivity. Results show that both human-likeness and performance remain similar across different $j$ values.
- **Requirement of high-quality human data (Reviewer HZYt, Reviewer rcnz)**: We trained MAQ with suboptimal human demonstrations. While success rates decreased, human-likeness metrics remained high, showing that MAQ does not necessarily require high-quality human data.
- **Robustness of our method to the amount of human demonstrations (Reviewer HJQk, Reviewer gwPt)**: We conducted an ablation study using different amounts of training data ratio in Door and Pen environments: 90% (as in the paper), 75%, 50%, and 25%, out of 25 human demonstration trajectories in total per task. Results show that the trajectory similarity scores remain consistent across different ratios. This shows that MAQ can still capture human-like behavior even with as few as 6 trajectories and demonstrates the robustness of our method.

We will incorporate these experiments and discussions into the final version. We thank the reviewers again for their time and insightful feedback in reviewing our paper.

---

### Decision · Program_Chairs · 2025-09-17

**Decision:**

Accept (poster)

**Comment:**

This paper addresses the challenge of developing reinforcement learning agents that exhibit human-like behavior while maintaining task performance. The authors formulate human-likeness as a trajectory optimization problem and propose Macro Action Quantization (MAQ), a framework that distills human demonstrations into discrete macro-actions using Conditional Vector Quantized Variational Autoencoders (VQVAE). The key innovation is learning fixed-length action sequences from human demonstrations and having RL agents operate in this macro-action space rather than primitive actions. The system is evaluated in four Adroit tasks (including opening door, hammering a nail, twirling a pen, moving a ball), where the generated behaviors are compared with human demonstrations using dynamic time warping and Wasserstein distances, success rate, as well as via human studies. Results show that the generated behaviors achieve higher similarity scores and success rates, while being less distinguishable from human demonstrations.

The main strengths for this work are as follows: (1)  The paper addresses an important but underexplored area in RL - achieving human-like behavior rather than just optimal performance. This is particularly relevant for applications requiring human-AI interaction and trust.
(2) The authors provide both quantitative trajectory similarity metrics and qualitative human studies. The inclusion of Turing tests and human ranking evaluations strengthens the validity of their human-likeness claims. (3) MAQ demonstrates good generalization across different RL algorithms (IQL, SAC, RLPD) and can be easily integrated into existing methods, making it practically useful. (4) The paper includes appropriate baselines, ablation studies, and addresses key concerns raised during review through additional experiments.

The main weaknesses of this work are as follows: (1) Experiments were confined to Adroit manipulation tasks. While the authors addressed this during rebuttal with Atari experiments, broader evaluation across diverse domains would strengthen generalizability claims.
(2) The approach relies on fixed-length action sequences (length 9), which may not capture the natural variability in human behavior segments. This constraint could limit applicability to tasks requiring variable-length behaviors. (3)  The paper lacks theoretical analysis of when and why the compositionality assumption of macro actions holds. The empirical success doesn't fully address the fundamental question of macro-action reusability. (4) While addressed in rebuttal, the additional computational costs of VQVAE training and inference deserve more thorough analysis in the main paper. (5) The submission misses analysis on critical hyperparameters including the minimum amount of human demonstrations required and the impact of codebook size K in the VQVAE framework, which was reported in the rebuttal. (6) There is a strong assumption of requiring human demonstrations specifically formatted as macro actions, which may severely limit the method's applicability and require domain-specific knowledge for macro action identification.

Reasons for acceptance: The work addresses an important gap: human-like behavior in RL is an underexplored but practically important problem. The combination of quantitative trajectory similarity improvements and positive human evaluation studies provides convincing evidence of the method's effectiveness. The technical approach is well-motivated, and the experimental methodology is appropriate, with good attention to potential confounds and alternative explanations. The framework's ability to integrate with existing RL algorithms and improve human-likeness across different methods demonstrates practical value. However, there are also a series of limitations in this version, including fixed-length macro actions, compositionality assumptions, and a lot of experiments are done in the rebuttal period instead of the submission, which affects the quality significantly. Although the thorough rebuttal addresses reviewer concerns, we can only accept this work after the necessary discussion and results are involved in the final version.